# How Does SimSiam Avoid Collapse Without Negative Samples? A Unified Understanding with Self-supervised Contrastive Learning

**Chaoning Zhang**\*& **Kang Zhang**\* & **Chenshuang Zhang** & **Trung X. Pham**
**Chang D. Yoo & In So Kweon**
Korea Advanced Institute of Science and Technology (KAIST), South Korea
chaoningzhang1990@gmail.com & zhangkang@kaist.ac.kr

## Abstract

To avoid collapse in self-supervised learning (SSL), a contrastive loss is widely used but often requires a large number of negative samples. Without negative samples yet achieving competitive performance, a recent work (Chen & He, 2021) has attracted significant attention for providing a minimalist simple Siamese (SimSiam) method to avoid collapse. However, the reason for how it avoids collapse without negative samples remains not fully clear and our investigation starts by revisiting the explanatory claims in the original SimSiam. After refuting their claims, we introduce vector decomposition for analyzing the collapse based on the gradient analysis of the $l_2$-normalized representation vector. This yields a unified perspective on how negative samples and SimSiam alleviate collapse. Such a unified perspective comes timely for understanding the recent progress in SSL.

## 1 Introduction

Beyond the success of NLP (Lan et al., 2020; Radford et al., 2019; Devlin et al., 2019; Su et al., 2020; Nie et al., 2020), self-supervised learning (SSL) has also shown its potential in the field of vision tasks (Li et al., 2021; Chen et al., 2021; El-Nouby et al., 2021). Without the ground-truth label, the core of most SSL methods lies in learning an encoder with augmentation-invariant representation (Bachman et al., 2019; He et al., 2020; Chen et al., 2020a; Caron et al., 2020; Grill et al., 2020). Specifically, they often minimize the representation distance between two positive samples, *i.e.* two augmented views of the same image, based on a Siamese network architecture (Bromley et al., 1993). It is widely known that for such Siamese networks there exists a degenerate solution, *i.e.* all outputs "collapsing" to an undesired constant (Chen et al., 2020a; Chen & He, 2021). Early works have attributed the collapse to lacking a repulsive component in the optimization goal and adopted contrastive learning (CL) with negative samples, *i.e.* views of different samples, to alleviate this problem. Introducing momentum into the target encoder, BYOL shows that Siamese architectures can be trained with only positive pairs. More recently, SimSiam (Chen & He, 2021) has caught great attention by further simplifying BYOL by removing the momentum encoder, which has been seen as a major milestone achievement in SSL for providing a *minimalist* method for achieving competitive performance. However, more investigation is required for the following question:

**How does SimSiam avoid collapse without negative samples?**

Our investigation starts with revisiting the explanatory claims in the original SimSiam paper (Chen & He, 2021). Notably, two components, *i.e.* stop gradient and predictor, are essential for the success of SimSiam (Chen & He, 2021). The reason has been mainly attributed to the stop gradient (Chen & He, 2021) by hypothesizing that it implicitly involves two sets of variables and SimSiam behaves like alternating between optimizing each set. Chen & He argue that the predictor $h$ is helpful in SimSiam because $h$ fills the gap to approximate expectation over augmentations (EOA).

Unfortunately, the above explanatory claims are found to be flawed due to reversing the two paths with and without gradient (see Sec. 2.2). This motivates us to find an alternative explanation, for which we introduce a simple yet intuitive framework for facilitating the analysis of collapse in SSL.

---

\*equal contribution

Specifically, we propose to decompose a representation vector into center and residual components. This decomposition facilitates understanding which gradient component is beneficial for avoiding collapse. Under this framework, we show that a basic Siamese architecture cannot prevent collapse, for which an extra gradient component needs to be introduced. With SimSiam interpreted as processing the optimization target with an inverse predictor, the analysis of its extra gradient shows that (a) its center vector helps prevent collapse via the de-centering effect; (b) its residual vector achieves dimensional de-correlation which also alleviates collapse.

Moreover, under the same gradient decomposition, we find that the extra gradient caused by negative samples in InfoNCE (He et al., 2019; Chen et al., 2020b;a; Tian et al., 2019; Khosla et al., 2020) also achieves de-centering and de-correlation in the same manner. It contributes to a unified understanding on various frameworks in SSL, which also inspires the investigation of hardness-awareness Wang & Liu (2021) from the inter-anchor perspective Zhang et al. (2022) for further bridging the gap between CL and non-CL frameworks in SSL. Finally, simplifying the predictor for more explainable SimSiam, we show that a single bias layer is sufficient for preventing collapse.

The basic experimental settings for our analysis are detailed in Appendix A.1 with a more specific setup discussed in the context. Overall, our work is the first attempt for performing a comprehensive study on how SimSiam avoids collapse without negative samples. Several works, however, have attempted to demystify the success of BYOL (Grill et al., 2020), a close variant of SimSiam. A technical report (Fetterman & Albrecht, 2020) has suggested the importance of batch normalization (BN) in BYOL for its success, however, a recent work (Richemond et al., 2020) refutes their claim by showing BYOL works without BN, which is discussed in Appendix B.

## 2 REVISITING SIMSIAM AND ITS EXPLANATORY CLAIMS

$l_2$-**normalized vector and optimization goal.** SSL trains an encoder $f$ for learning discriminative representation and we denote such representation as a vector $\boldsymbol{z}$, *i.e.* $f(x) = \boldsymbol{z}$ where $x$ is a certain input. For the augmentation-invariant representation, a straightforward goal is to minimize the distance between the representations of two positive samples, *i.e.* augmented views of the same image, for which mean squared error (MSE) is a default choice. To avoid scale ambiguity, the vectors are often $l_2$-normalized, *i.e.* $\boldsymbol{Z} = \boldsymbol{z}/||\boldsymbol{z}||$ (Chen & He, 2021), before calculating the MSE:

$$\mathcal{L}_{MSE} = (\boldsymbol{Z}_a - \boldsymbol{Z}_b)^2/2 - 1 = -\boldsymbol{Z}_a \cdot \boldsymbol{Z}_b = L_{cosine}, \tag{1}$$

which shows the equivalence of a normalized MSE loss to the cosine loss (Grill et al., 2020).

**Collapse in SSL and solution of SimSiam.** Based on a Siamese architecture, the loss in Eq 1 causes the collapse, i.e. $f$ always outputs a constant regardless of the input variance. We refer to this Siamese architecture with loss Eq 1 as *Naive Siamese* in the remainder of paper. Contrastive loss with negative samples is a widely used solution (Chen et al., 2020a). Without using negative samples, SimSiam solves the collapse problem via predictor and stop gradient, based on which the encoder is optimized with a symmetric loss:

$$L_{SimSiam} = -(\boldsymbol{P}_a \cdot \text{sg}(\boldsymbol{Z}_b) + \boldsymbol{P}_b \cdot \text{sg}(\boldsymbol{Z}_a)), \tag{2}$$

where $\text{sg}(\cdot)$ is *stop gradient* and $\boldsymbol{P}$ is the output of predictor $h$, *i.e.* $\boldsymbol{p} = h(\boldsymbol{z})$ and $\boldsymbol{P} = \boldsymbol{p}/||\boldsymbol{p}||$.

### 2.1 REVISING EXPLANATORY CLAIMS IN SIMSIAM

**Interpreting stop gradient as AO.** Chen & He hypothesize that the stop gradient in Eq 2 is an implementation of Alternating between the Optimization of two sub-problems, which is denoted as AO. Specifically, with the loss considered as $\mathcal{L}(\theta, \eta) = \mathbb{E}_{x,\mathcal{T}}\left[\left\|\mathcal{F}_\theta(\mathcal{T}(x)) - \eta_x\right\|^2\right]$, the optimization objective $\min_{\theta, \eta} \mathcal{L}(\theta, \eta)$ can be solved by alternating $\eta^t \leftarrow \arg\min_\eta \mathcal{L}(\theta^t, \eta)$ and $\theta^{t+1} \leftarrow \arg\min_\theta \mathcal{L}(\theta, \eta^t)$. It is acknowledged that this hypothesis does not fully explain why the collapse is prevented (Chen & He, 2021). Nonetheless, they mainly attribute SimSiam success to the stop gradient with the interpretation that AO might make it difficult to approach a constant $\forall x$.

**Interpreting predictor as EOA.** The AO problem (Chen & He, 2021) is formulated independent of predictor $h$, for which they believe that the usage of predictor $h$ is related to approximating EOA for filling the gap of ignoring $\mathbb{E}_\mathcal{T}[\cdot]$ in a sub-problem of AO. The approximation of $\mathbb{E}_\mathcal{T}[\cdot]$ is summarized

in Appendix A.2. Chen & He support their interpretation by proof-of-concept experiments. Specifically, they show that updating $\eta_x^t$ with a moving-average $\eta_x^t \leftarrow m * \eta_x^t + (1 - m) * \mathcal{F}_{\theta^t}(\mathcal{T}'(x))$ can help prevent collapse without predictor (see Fig. 1 (b)). Given that the training completely fails when the predictor and moving average are both removed, at first sight, their reasoning seems valid.

## 2.2 DOES THE PREDICTOR FILL THE GAP TO APPROXIMATE EOA?

**Reasoning flaw.** Considering the stop gradient, we divide the framework into two sub-models with different paths and term them Gradient Path (GP) and Stop Gradient Path (SGP). For SimSiam, only the sub-model with GP includes the predictor (see Fig. 1 (a)). We point out that their reasoning flaw of predictor analysis lies in the *reverse of GP and SGP*. By default, the moving-average sub-model, as shown in Fig. 1 (b), is on the same side as SGP. Note that Fig. 1 (b) is conceptually similar to Fig. 1 (c) instead of Fig. 1 (a). It is worth mentioning that the Mirror SimSiam in Fig. 1 (c) is what stop gradient in the original SimSiam avoids. Therefore, it is problematic to perceive $h$ as EOA.

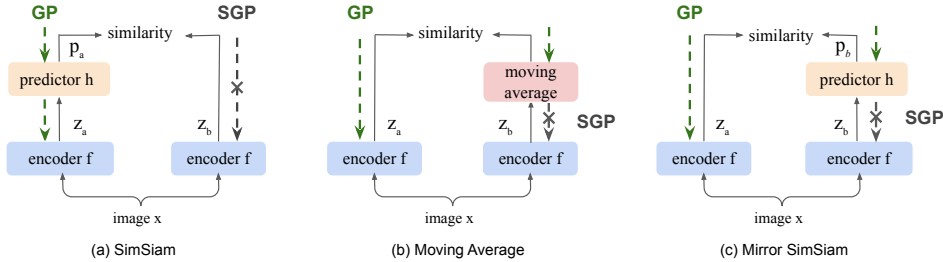

Figure 1: **Reasoning Flaw in SimSiam.** (a) Standard SimSiam architecture. (b) Moving-Average Model proposed in the proof-of-concept experiment (Chen & He, 2021). (c) Mirror SimSiam, which has the same model architecture as SimSiam but with the reverse of GP and SGP.

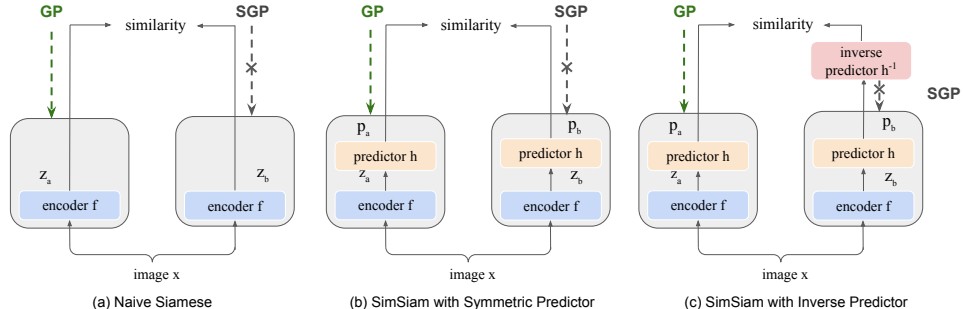

Figure 2: Different architectures of Siamese model. When it is trained experimentally, the inverse predictor in (c) has the same architecture as predictor $h$.

| Method | # aug | Collapse | Std | Top-1 (%) |
|---|---|---|---|---|
| Moving average | 2 | × | 0.0108 | 46.57 |
| Same batch | 10 | ✓ | 0 | 1 |
| Same batch | 25 | ✓ | 0 | 1 |

Table 1: Influence of Explicit EOA. Detailed setup is reported in Appendix A.3

**Explicit EOA does not prevent collapse.** (Chen & He, 2021) points out that "in practice, it would be unrealistic to actually compute the expectation $\mathbb{E}_{\mathcal{T}}[\cdot]$. But it may be possible for a neural network (e.g., the preditor $h$) to learn to predict the expectation, while the sampling of $\mathcal{T}$ is implicitly distributed across multiple epochs." If implicitly sampling across multiple epochs is beneficial, explicitly sampling sufficient large $N$ augmentations in a batch with the latest model would be more beneficial to approximate $\mathbb{E}_{\mathcal{T}}[\cdot]$. However, Table 1 shows that the collapse still occurs and suggests that the equivalence between predictor and EOA does not hold.

## 2.3 Asymmetric interpretation of predictor with stop gradient in SimSiam

**Symmetric Predictor does not prevent collapse.** The difference between Naive Siamese and Simsiam lies in whether the gradient in backward propagation flows through a predictor, however, we show that this propagation helps avoid collapse only when the predictor is not included in the SGP path. With $h$ being trained the same as Eq 2, we optimize the encoder $f$ through replacing the $\boldsymbol{Z}$ in Eq 2 with $\boldsymbol{P}$. The results in Table. 2 show that it still leads to collapse. Actually, this is well expected by perceiving $h$ to be part of the new encoder $F$, *i.e.* $\boldsymbol{p} = F(x) = h(f(x))$. In other words, the symmetric architectures *with* and *without* predictor $h$ both lead to collapse.

**Predictor with stop gradient is asymmetric.** Clearly, how SimSiam avoids collapse lies in its asymmetric architecture, *i.e.* one path with $h$ and the other without $h$. Under this asymmetric architecture, the role of stop gradient is to only allow the path with predictor to be optimized with the encoder output as the target, not vice versa. In other words, the SimSiam avoids collapse by excluding Mirror SimSiam (Fig. 1 (c)) which has a loss (mirror-like Eq 2) as $\mathcal{L}_{\text{Mirror}} = -(\boldsymbol{P}_a \cdot \boldsymbol{Z}_b + \boldsymbol{P}_b \cdot \boldsymbol{Z}_a)$, where stop gradient is put on the input of $h$, *i.e.* $\boldsymbol{p}_a = h(\text{sg}[\boldsymbol{z}_a])$ and $\boldsymbol{p}_b = h(\text{sg}[\boldsymbol{z}_b])$.

**Predictor *vs*. inverse predictor.** We interpret $h$ as a function mapping from $\boldsymbol{z}$ to $\boldsymbol{p}$, and introduce a conceptual inverse mapping $h^{-1}$, *i.e.* $\boldsymbol{z} = h^{-1}(\boldsymbol{p})$. Here, as shown in Table 2, SimSiam with symmetric predictor (Fig. 2 (b)) leads to collapse, while SimSiam (Fig. 1 (a)) avoids collapse. With the conceptual $h^{-1}$, we interpret Fig. 1 (a) the same as Fig. 2 (c) which differs from Fig. 2 (b) via changing the optimization target from $\boldsymbol{p}_b$ to $\boldsymbol{z}_b$, *i.e.* $\boldsymbol{z}_b = h^{-1}(\boldsymbol{p}_b)$. This interpretation

| Method | Collapse | Top-1 (%) |
|---|---|---|
| Simsiam | ✗ | 66.62 |
| Mirror SimSiam | ✓ | 1 |
| Naive Siamese | ✓ | 1 |
| Symmetric Predictor | ✓ | 1 |

Table 2: Results of various Siamese architectures. Detailed trend and setup are reported in Appendix A.4

suggests that the collapse can be avoided by processing the optimization target with $h^{-1}$. By contrast, Fig. 1 (c) and Fig. 2 (a) both lead to collapse, suggesting that processing the optimization target with $h$ is *not* beneficial for preventing collapse. Overall, asymmetry alone does not guarantee collapse avoidance, which requires the optimization target to be processed by $h^{-1}$ not $h$.

**Trainable inverse predictor and its implication on EOA.** In the above, we propose a *conceptual* inverse predictor $h^{-1}$ in Fig. 2 (c), however, it remains yet unknown whether such an inverse predictor is *experimentally trainable*. A detailed setup for this investigation is reported in Appendix A.5. The results in Fig. 3 show that a learnable $h^{-1}$ leads to slightly inferior performance, which is expected because $h^{-1}$ cannot make the trainable inverse predictor output $\boldsymbol{z}_b^*$ completely the same as $\boldsymbol{z}_b$. Note that it would be equivalent to SimSiam if $\boldsymbol{z}_b^* = \boldsymbol{z}_b$. Despite a slight performance drop, the results confirm that $h^{-1}$ is trainable. The fact that $h^{-1}$ is trainable provides additional evidence that the role $h$ plays in SimSiam is not EOA

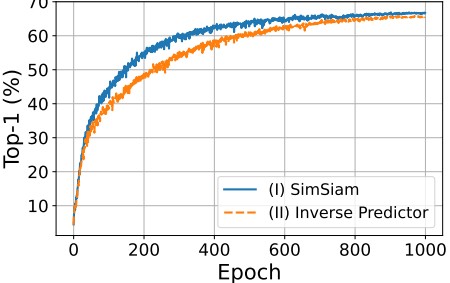

Figure 3: Comparison of original SimSiam and SimSiam with Inverse Predictor.

because theoretically $h^{-1}$ cannot restore a random augmentation $\mathcal{T}'$ from an expectation $\boldsymbol{p}$, where $\boldsymbol{p} = h(\boldsymbol{z}) = \mathbb{E}_{\mathcal{T}}\left[\mathcal{F}_{\theta^t}(\mathcal{T}(x))\right]$.

## 3 Vector decomposition for understanding collapse

By default, InfoNCE (Chen et al., 2020a) and SimSiam (Chen & He, 2021) both adopt $l_2$-normalization in their loss for avoiding scale ambiguity. We treat the $l_2$-normalized vector, *i.e.* $\boldsymbol{Z}$, as the encoder output, which significantly simplifies gradient derivation and the following analysis.

**Vector decomposition.** For the purpose of analysis, we propose to decompose $\boldsymbol{Z}$ into two parts, $\boldsymbol{Z} = \boldsymbol{o} + \boldsymbol{r}$, where $\boldsymbol{o}$, $\boldsymbol{r}$ denote *center vector* and *residual vector* respectively. Specifically, the center vector $\boldsymbol{o}$ is defined as an average of $\boldsymbol{Z}$ over the whole representation space $\boldsymbol{o}_z = \mathbb{E}[\boldsymbol{Z}]$. However, we approximate it with all vectors in current mini-batch, *i.e.* $\boldsymbol{o}_z = \frac{1}{M}\sum_{m=1}^{M}\boldsymbol{Z}_m$, where $M$ is the mini-batch size. We define the residual vector $\boldsymbol{r}$ as the residual part of $\boldsymbol{Z}$, *i.e.* $\boldsymbol{r} = \boldsymbol{Z} - \boldsymbol{o}_z$.

### 3.1 Collapse from the vector perspective

**Collapse: from result to cause.** A Naive Siamese is well expected to collapse since the loss is designed to minimize the distance between positive samples, for which a constant constitutes an optimal solution to minimize such loss. When the collapse occurs, $\forall i, \boldsymbol{Z}_i = \frac{1}{M} \sum_{m=1}^{M} \boldsymbol{Z}_m = \boldsymbol{o}_z$, where $i$ denotes a random sample index, which shows the constant vector is $\boldsymbol{o}_z$ in this case. This interpretation only suggests a possibility that a dominant $\boldsymbol{o}$ can be one of the viable solutions, while the optimization, such as SimSiam, might still lead to a non-collapse solution. This merely describes $\boldsymbol{o}$ as the *consequence* of the collapse, and our work investigates the *cause* of such collapse through analyzing the influence of individual gradient components, *i.e.* $\boldsymbol{o}$ and $\boldsymbol{r}$ during training.

**Competition between $\boldsymbol{o}$ and $\boldsymbol{r}$.** Complementary to the Standard Deviation (Std) (Chen & He, 2021), for indicating collapse, we introduce the ratio of $\boldsymbol{o}$ in $\boldsymbol{z}$, *i.e.* $m_o = ||\boldsymbol{o}||/||\boldsymbol{z}||$, where $|| * ||$ is the $L_2$ norm. Similarly, the ratio of $\boldsymbol{r}$ in $\boldsymbol{z}$ is defined as $m_r = ||\boldsymbol{r}||/||\boldsymbol{z}||$. When collapse happens, *i.e.* all vectors $\boldsymbol{Z}$ are close to the center vector $\boldsymbol{o}$, $m_o$ approaches 1 and $m_r$ approaches 0, which is not desirable for SSL. A desirable case would be a relatively small $m_o$ and a relatively large $m_r$, suggesting a relatively small (large) contribution of $\boldsymbol{o}$ ($\boldsymbol{r}$) in each $\boldsymbol{Z}$. We interpret the cause of collapse as a competition between $\boldsymbol{o}$ and $\boldsymbol{r}$ where $\boldsymbol{o}$ dominates over $\boldsymbol{r}$, *i.e.* $m_o \gg m_r$. For Eq 1, the derived negative gradient on $\boldsymbol{Z}_a$ (ignoring $\boldsymbol{Z}_b$ for simplicity due to symmetry) is shown as:

$$\mathcal{G}_{cosine} = -\frac{\partial \mathcal{L}_{MSE}}{\partial \boldsymbol{Z}_a} = \boldsymbol{Z}_b - \boldsymbol{Z}_a \iff -\frac{\partial \mathcal{L}_{cosine}}{\partial \boldsymbol{Z}_a} = \boldsymbol{Z}_b, \tag{3}$$

where the gradient component $\boldsymbol{Z}_a$ is a *dummy* term because the loss $-\boldsymbol{Z}_a \cdot \boldsymbol{Z}_a = -1$ is a constant having zero gradient on the encoder $f$.

**Conjecture1.** With $\boldsymbol{Z}_a = \boldsymbol{o}_z + \boldsymbol{r}_a$, we conjecture that the gradient component of $\boldsymbol{o}_z$ is expected to update the encoder to boost the center vector thus increase $m_o$, while the gradient component of $\boldsymbol{r}_a$ is expected to behave in the opposite direction to increase $m_r$. A random gradient component is expected to have a relatively small influence.

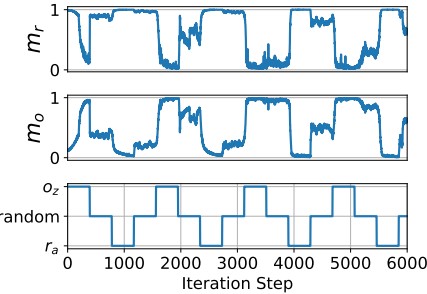

Figure 4: Influence of various gradient components on $m_r$ and $m_o$.

To verify the above conjecture, we revisit the *dummy* gradient term $\boldsymbol{Z}_a$. We design loss $-\boldsymbol{Z}_a \cdot \text{sg}(\boldsymbol{o}_z)$ and $-\boldsymbol{Z}_a \cdot \text{sg}(\boldsymbol{Z}_a - \boldsymbol{o}_z)$ to show the influence of gradient component $\boldsymbol{o}$ and $\boldsymbol{r}_a$ respectively. The results in Fig. 4 show that the gradient component $\boldsymbol{o}_z$ has the effect of increasing $m_o$ while decreasing $m_r$. On the contrary, $\boldsymbol{r}_a$ helps increase $m_r$ while decreasing $m_o$. Overall, the results verify Conjecture1.

### 3.2 Extra gradient component for alleviating collapse

**Revisit collapse in a symmetric architecture.** Based on Conjecture1, here, we provide an intuitive interpretation on why a symmetric Siamese architecture, such as Fig. 2 (a) and (b), cannot be trained without collapse. Take Fig. 2 (a) as example, the gradient in Eq 3 can be interpreted as two equivalent forms, from which we choose $\boldsymbol{Z}_b - \boldsymbol{Z}_a = (\boldsymbol{o}_z + \boldsymbol{r}_b) - (\boldsymbol{o}_z + \boldsymbol{r}_a) = \boldsymbol{r}_b - \boldsymbol{r}_a$. Since $\boldsymbol{r}_b$ comes from the same positive sample as $\boldsymbol{r}_a$, it is expected that $\boldsymbol{r}_b$ also increases $m_r$, however, this effect is expected to be smaller than that of $\boldsymbol{r}_a$, thus causing collapse.

**Basic gradient and Extra gradient components.** The negative gradient on $\boldsymbol{Z}_a$ in Fig. 2 (a) is derived as $\boldsymbol{Z}_b$, while that on $\boldsymbol{P}_a$ in Fig. 2 (b) is derived as $\boldsymbol{P}_b$. We perceive $\boldsymbol{Z}_b$ and $\boldsymbol{P}_b$ in these basic Siamese architectures as the *Basic Gradient*. Our above interpretation shows that such basic components cannot prevent collapse, for which an **Extra G**radient component, denoted as $\boldsymbol{G}_e$, needs to be introduced to break the **symmetry**. As the term suggests, $\boldsymbol{G}_e$ is defined as a gradient term that is relative to the basic gradient in a basic Siamese architecture. For example, negative samples can be introduced to Naive Siamese (Fig. 2 (a)) for preventing collapse, where the extra gradient caused by negative samples can thus be perceived as $\boldsymbol{G}_e$ with $\boldsymbol{Z}_b$ as the basic gradient. Similarly, we can also disentangle the negative gradient on $\boldsymbol{P}_a$ in SimSiam (Fig. 1 (a)), *i.e.* $\boldsymbol{Z}_b$, into a basic gradient (which is $\boldsymbol{P}_b$) and $\boldsymbol{G}_e$ which is derived as $\boldsymbol{Z}_b - \boldsymbol{P}_b$ (note that $\boldsymbol{Z}_b = \boldsymbol{P}_b + \boldsymbol{G}_e$). We analyze how $\boldsymbol{G}_e$ prevents collapse via studying the independent roles of its center vector $\boldsymbol{o}_e$ and residual vector $\boldsymbol{r}_e$.

### 3.3 A TOY EXAMPLE EXPERIMENT WITH NEGATIVE SAMPLE

**Which repulsive component helps avoid collapse?** Existing works often attribute the collapse in Naive Siamese to lacking a *repulsive part* during the optimization. This explanation has motivated previous works to adopt contrastive learning, *i.e.* attracting the positive samples while *repulsing* the negative samples. We experiment with a simple triplet loss[1], $\mathcal{L}_{tri} = -\boldsymbol{Z}_a \cdot \text{sg}(\boldsymbol{Z}_b - \boldsymbol{Z}_n)$, where $\boldsymbol{Z}_n$ indicates the representation of a **N**egative sample. The derived negative gradient on $\boldsymbol{Z}_a$ is $\boldsymbol{Z}_b - \boldsymbol{Z}_n$, where $\boldsymbol{Z}_b$ is the basic gradient component and thus $\boldsymbol{G}_e = -\boldsymbol{Z}_n$ in this setup. For a sample representation, what determines it as a positive sample for attracting or a negative sample for repulsing is the residual component, *thus it might be tempting to interpret that $\boldsymbol{r}_e$ is the key component of repulsive part that avoids the collapse*. However, the results in Table 3 show that the component beneficial for preventing collapse inside $\boldsymbol{G}_e$ is $\boldsymbol{o}_e$ instead of $\boldsymbol{r}_e$. Specifically, to explore the individual influence of $\boldsymbol{o}_e$ and $\boldsymbol{r}_e$ in the $\boldsymbol{G}_e$, we design two experiments by removing one component while keeping the other one. In the first experiment, we remove the $\boldsymbol{r}_e$ in $\boldsymbol{G}_e$ while keeping the $\boldsymbol{o}_e$. By contrast, the $\boldsymbol{o}_e$ is removed while keeping the $\boldsymbol{r}_e$ in the second experiment. In contrast to what existing explanations may expect, we find that the residual component $\boldsymbol{o}_e$ prevents collapses. With Conjecture1, a gradient component alleviates collapse if it has negative center vector. In this setup, $\boldsymbol{o}_e = -\boldsymbol{o}_z$, thus $\boldsymbol{o}_e$ has the de-centering role for preventing collapse. On the contrary, $\boldsymbol{r}_e$ does not prevent collapse and keeping $\boldsymbol{r}_e$ even decreases the performance ($36.21\% < 47.41\%$). Since the negative sample is randomly chosen, $\boldsymbol{r}_e$ just behaves like a random noise on the optimization to decrease performance.

| Method | $\mathcal{L}_{triplet}$ | Std | $m_o$ | $m_r$ | Collapse | Top-1 (%) |
|---|---|---|---|---|---|---|
| Baseline | $-\boldsymbol{Z}_a \cdot \text{sg}(\boldsymbol{Z}_b + \boldsymbol{G}_e)$ | 0.020 | 0.026 | 0.99 | $\times$ | 36.21 |
| Removing $\boldsymbol{r}_e$ | $-\boldsymbol{Z}_a \cdot \text{sg}(\boldsymbol{Z}_b + \boldsymbol{o}_e)$ | 0.02005 | 0.026 | 0.99 | $\times$ | 47.41 |
| Removing $\boldsymbol{o}_e$ | $-\boldsymbol{Z}_a \cdot \text{sg}(\boldsymbol{Z}_b + \boldsymbol{r}_e)$ | 0 | 1 | 0 | $\checkmark$ | 1 |

Table 3: Gradient component analysis with a random negative sample.

### 3.4 DECOMPOSED GRADIENT ANALYSIS IN SIMSIAM

It is challenging to derive the gradient on the encoder output in SimSiam due to a nonlinear MLP module in $h$. The negative gradient on $\boldsymbol{P}_a$ for $\mathcal{L}_{SimSiam}$ in Eq 2 can be derived as

$$\mathcal{G}_{SimSiam} = -\frac{\partial \mathcal{L}_{SimSiam}}{\partial \boldsymbol{P}_a} = \boldsymbol{Z}_b = \boldsymbol{P}_b + (\boldsymbol{Z}_b - \boldsymbol{P}_b) = \boldsymbol{P}_b + \boldsymbol{G}_e, \quad (4)$$

where $\boldsymbol{G}_e$ indicates the aforementioned extra gradient component. To investigate the influence of $\boldsymbol{o}_e$ and $\boldsymbol{r}_e$ on the collapse, similar to the analysis with the toy example experiment in Sec. 3.3, we design the experiment by removing one component while keeping the other. The results are reported in Table 4. As expected, the model collapses when both components in $\boldsymbol{G}_e$ are removed and the best performance is achieved when both components are kept. Interestingly, the model does not collapse when either $\boldsymbol{o}_e$ or $\boldsymbol{r}_e$ is kept. To start, we analyze how $\boldsymbol{o}_e$ affects the collapse based on Conjecture1.

| $\boldsymbol{o}_e$ | $\boldsymbol{r}_e$ | Collapse | Top-1 (%) |
|---|---|---|---|
| $\checkmark$ | $\checkmark$ | $\times$ | 66.62 |
| $\checkmark$ | $\times$ | $\times$ | 48.08 |
| $\times$ | $\checkmark$ | $\times$ | 66.15 |
| $\times$ | $\times$ | $\checkmark$ | 1 |

Table 4: Gradient component analysis for SimSiam.

**How $\boldsymbol{o}_e$ alleviates collapse in SimSiam.** Here, $\boldsymbol{o}_p$ is used to denote the center vector of $\boldsymbol{P}$ to differentiate from the above introduced $\boldsymbol{o}_z$ for denoting that of $Z$. In this setup $\boldsymbol{G}_e = \boldsymbol{Z}_b - \boldsymbol{P}_b$, thus the residual gradient component is derived to be $\boldsymbol{o}_e = \boldsymbol{o}_z - \boldsymbol{o}_p$. *With Conjecture1, it is well expected that $\boldsymbol{o}_e$ helps prevent collapse if $\boldsymbol{o}_e$ contains negative $\boldsymbol{o}_p$ since the analyzed vector is $\boldsymbol{P}_a$.* To determine the amount of component of $\boldsymbol{o}_p$ existing in $\boldsymbol{o}_e$, we measure the cosine similarity between $\boldsymbol{o}_e - \eta_p \boldsymbol{o}_p$ and $\boldsymbol{o}_p$ for a wide range of $\eta_p$. The results in Fig. 5 (a) show that their cosine similarity is zero when $\eta_p$ is around $-0.5$, suggesting $\boldsymbol{o}_e$ has $\approx -0.5\boldsymbol{o}_p$. With Conjecture1, this negative $\eta_p$ explains why SimSiam avoids collapse from the perspective of de-centering.

**How $\boldsymbol{o}_e$ causes collapse in Mirror SimSiam.** As mentioned above, the collapse occurs in Mirror SimSiam, which can also be explained by analyzing its $\boldsymbol{o}_e$. Here, $\boldsymbol{o}_e = \boldsymbol{o}_p - \boldsymbol{o}_z$, for which we evaluate the amount of component $\boldsymbol{o}_z$ existing in $\boldsymbol{o}_e$ via reporting the similarity between $\boldsymbol{o}_e - \eta_z \boldsymbol{o}_z$

---

[1]Note that the triplet loss here does not have clipping form as in Schroff et al. (2015) for simplicity.

and $\boldsymbol{o}_z$. The results in Fig. 5 (a) show that their cosine similarity is zero when $\eta_z$ is set to around 0.2. This positive $\eta_z$ explains why Fig. 1(c) causes collapse from the perspective of de-centering.

Overall, we find that processing the optimization target with $h^{-1}$, as in Fig. 2 (c), alleviates collapse ($\eta_p \approx -0.5$), while processing it with $h$, as in Fig. 1(c), actually strengthens the collapse ($\eta_z \approx 0.2$). In other words, via the analysis of $\boldsymbol{o}_e$, our results help explain how SimSiam avoids collapse as well as how Mirror SimSiam causes collapse from a straightforward de-centering perspective.

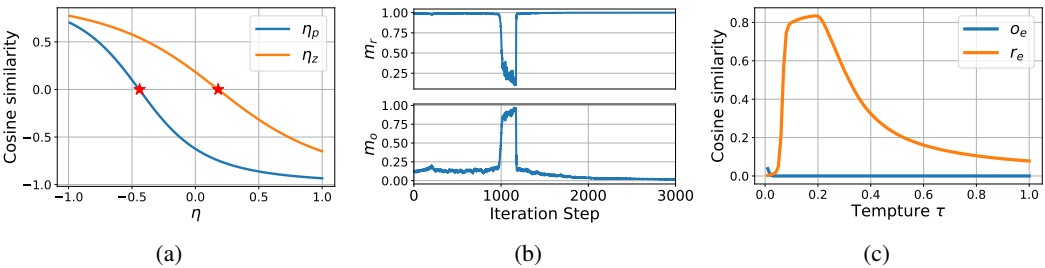

(a)                                    (b)                                    (c)

Figure 5: (a) Investigating the amount of $\boldsymbol{o}_p$ existing in $\boldsymbol{o}_z - \boldsymbol{o}_p$ and the amount of $\boldsymbol{o}_z$ existing in $\boldsymbol{o}_p - \boldsymbol{o}_z$. (b) Normally train the model as SimSiam for 5 epochs, then using collapsing loss for 1 epoch to reduce $m_r$, followed by a correlation regularization loss. (c) Cosine similarity between $\boldsymbol{r}_e$ ($\boldsymbol{o}_e$) and gradient on $\boldsymbol{Z}_a$ induced by a correlation regularization loss.

**Relation to prior works.** Motivated from preventing the collapse to a constant, multiple prior works, such as W-MSE (Ermolov et al., 2021), Barlow-twins (Zbontar et al., 2021), DINO (Caron et al., 2021), explicitly adopt de-centering to prevent collapse. Despite various motivations, we find that they all implicitly introduce an $\boldsymbol{o}_e$ that contains a negative center vector. The success of their approaches aligns well with our Conjecture1 as well as our above empirical results. Based on our findings, we argue that the effect of de-centering can be perceived as $\boldsymbol{o}_e$ having a negative center vector. With this interpretation, we are the first to demonstrate that how SimSiam with predictor and stop gradient avoids collapse can be explained from the perspective of de-centering.

**Beyond de-centering for avoiding collapse.** In the toy example experiment in Sec. 3.3, $\boldsymbol{r}_e$ is found to be *not* beneficial for preventing collapse and keeping $\boldsymbol{r}_e$ even decreases the performance. Interestingly, as shown in Table 4, we find that $\boldsymbol{r}_e$ alone is sufficient for preventing collapse and achieves comparable performance as $\boldsymbol{G}_e$. This can be explained from the perspective of dimensional de-correlation, which will be discussed in Sec. 3.5.

### 3.5 DIMENSIONAL DE-CORRELATION HELPS PREVENT COLLAPSE

**Conjecture2 and motivation.** We conjecture that dimensional de-correlation increases $m_r$ for preventing collapse. The motivation is straightforward as follows. The dimensional correlation would be minimum if only a single dimension has a very high value for every individual class and the dimension changes for different classes. In another extreme case, when all the dimensions have the same values, equivalent to having a single dimension, which already collapses by itself in the sense of losing representation capacity. Conceptually, $\boldsymbol{r}_e$ has no direct influence on the center vector, thus we interpret that $\boldsymbol{r}_e$ prevents collapse through increasing $m_r$.

To verify the above conjecture, we train SimSiam normally with the loss in Eq 2 and train for several epochs with the loss in Eq 1 for intentionally decreasing the $m_r$ to close to zero. Then, we train the loss with only a correlation regularization term, which is detailed in Appendix A.6. The results in Fig. 5 (b) show that this regularization term increases $m_r$ at a very fast rate.

**Dimensional de-correlation in SimSiam.** Assuming $h$ only has a single FC layer to exclude the influence of $\boldsymbol{o}_e$, the weights in FC are expected to learn the correlation between different dimensions for the encoder output. This interpretation echos well with the finding that the eigenspace of $h$ weight aligns well with that of correlation matrix (Tian et al., 2021). In essence, the $h$ is trained to minimize the cosine similarity between $h(\boldsymbol{z}_a)$ and $I(\boldsymbol{z}_b)$, where $I$ is identity mapping. Thus, $h$ that learns the correlation is optimized close to $I$, which is conceptually equivalent to optimizing with the goal of de-correlation for $\boldsymbol{Z}$. As shown in Table 4, for SimSiam, $\boldsymbol{r}_e$ alone also prevents collapse, which

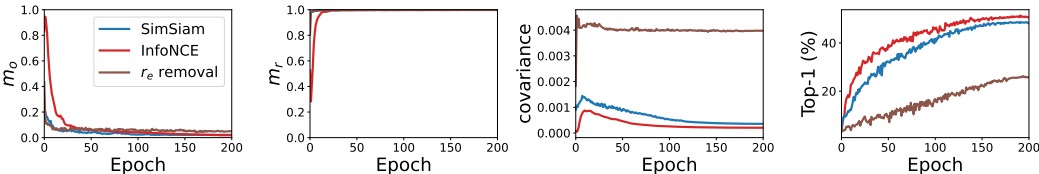

Figure 6: Influence of various gradient components on $m_r$ and $m_o$.

is attributed to the de-correlation effect since $\boldsymbol{r}_e$ has no de-centering effect. We observe from Fig. 6 that except in the first few epochs, SimSiam decreases the covariance during the whole training. Fig. 6 also reports the results for InfoNCE which will be discussed in Sec. 4.

# 4 TOWARDS A UNIFIED UNDERSTANDING OF RECENT PROGRESS IN SSL

**De-centering and de-correlation in InfoNCE.** InfoNCE loss is a default choice in multiple seminal contrastive learning frameworks (Sohn, 2016; Wu et al., 2018; Oord et al., 2018; Wang & Liu, 2021). The derived negative gradient of InfoNCE on $\boldsymbol{Z}_a$ is proportional to $\boldsymbol{Z}_b + \sum_{i=0}^{N} -\lambda_i \boldsymbol{Z}_i$, where $\lambda_i = \frac{\exp(\boldsymbol{Z}_a \cdot \boldsymbol{Z}_i / \tau)}{\sum_{i=0}^{N} \exp(\boldsymbol{Z}_a \cdot \boldsymbol{Z}_i / \tau)}$, and $\boldsymbol{Z}_0 = \boldsymbol{Z}_b$ for notation simplicity. See Appendix A.7 for the detailed derivation. The extra gradient component $\boldsymbol{G}_e = \sum_{i=0}^{N} -\lambda_i \boldsymbol{Z}_i = -\boldsymbol{o}_z - \sum_{i=0}^{N} \lambda_i \boldsymbol{r}_i$, for which $\boldsymbol{o}_e = -\boldsymbol{o}_z$ and $\boldsymbol{r}_e = -\sum_{i=0}^{N} \lambda_i \boldsymbol{r}_i$. Clearly, $\boldsymbol{o}_e$ contains negative $\boldsymbol{o}_z$ as de-centering for avoiding collapse, which is equivalent to the toy example in Sec. 3.3 when the $\boldsymbol{r}_e$ is removed. Regarding $\boldsymbol{r}_e$, the main difference between $\mathcal{L}_{tri}$ in the toy example and InfoNCE is that the latter exploits a batch of negative samples instead of a random one. $\lambda_i$ is proportional to $\exp(\boldsymbol{Z}_a \cdot \boldsymbol{Z}_i)$, indicating that a large weight is put on the negative sample when it is more similar to the anchor $\boldsymbol{Z}_a$, for which, intuitively, its dimensional values tend to have a high correlation with $\boldsymbol{Z}_a$. Thus, $\boldsymbol{r}_e$ containing such negative representation with a high weight tends to decrease dimensional correlation. To verify this intuition, we measure the cosine similarity between $\boldsymbol{r}_e$ and the gradient on $\boldsymbol{Z}_a$ induced by a correlation regularization loss. The results in Fig. 5 (c) show that their gradient similarity is high for a wide range of temperature values, especially when $\tau$ is around 0.1 or 0.2, suggesting $\boldsymbol{r}_e$ achieves similar role as an explicit regularization loss for performing de-correlation. Replacing $\boldsymbol{r}_e$ with $\boldsymbol{o}_e$ leads to a low cosine similarity, which is expected because $\boldsymbol{o}_e$ has no de-correlation effect.

The results of InfoNCE in Fig. 6 resembles that of SimSiam in terms of the overall trend. For example, InfoNCE also decreases the covariance value during training. Moreover, we also report the results of InfoNCE where $\boldsymbol{r}_e$ is removed for excluding the de-correlation effect. Removing $\boldsymbol{r}_e$ from the InfoNCE loss leads to a high covariance value during the whole training. Removing $\boldsymbol{r}_e$ also leads to a significant performance drop, which echos with the finding in (Bardes et al., 2021) that dimensional de-correlation is essential for competitive performance. Regarding how $\boldsymbol{r}_e$ in InfoNCE achieves de-correlation, formally, we **hypothesize** that *the de-correlation effect in InfoNCE arises from the biased weights ($\lambda_i$) on negative samples.* This hypothesis is corroborated by the temperature analysis in Fig. 7. We find that a higher temperature makes the weight distribution of $\lambda_i$ more balanced indicated a higher entropy of $\lambda_i$, which echos with the finding in (Wang & Liu, 2021). Moreover, we observe that a higher temperature also tends to increase the covariance value. Overall, with temperature as the control variable, we find that more balanced weights among negative samples decrease the de-correlation effect, which constitutes an evidence for our hypothesis.

**Unifying SimSiam and InfoNCE.** At first sight, there is no conceptual similarity between SimSiam and InfoNCE, and this is why the community is intrigued by the success of SimSiam without negative samples. Through decomposing the $\boldsymbol{G}_e$ into $\boldsymbol{o}_e$ and $\boldsymbol{r}_e$, we find that for both, their $\boldsymbol{o}_e$ plays the role of de-centering and their $\boldsymbol{r}_e$ behaves like de-correlation. In this sense, we bring two seemingly irrelevant frameworks into a unified perspective with disentangled de-centering and de-correlation.

**Beyond SimSiam and InfoNCE.** In SSL, there is a trend of performing *explicit* manipulation of de-centering and de-correlation, for which W-MSE (Ermolov et al., 2021), Barlow-twins (Zbontar et al., 2021), DINO (Caron et al., 2021) are three representative works. They often achieve performance comparable to those with InfoNCE or SimSiam. Towards a unified understanding of recent progress in SSL, our work is most similar to a concurrent work (Bardes et al., 2021). Their work is mainly inspired by Barlow-twins (Zbontar et al., 2021) but decomposes its loss into three explicit components. By contrast, our work is motivated to answer the question of how SimSiam prevents

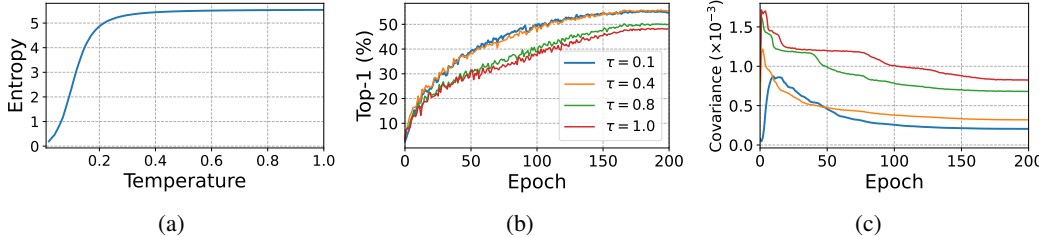

Figure 7: Influence of temperature. (a) Entropy of $\lambda_i$ with regard to temperature; (b) Top-1 accuracy trend with various temperature; (c) Covariance trend with various temperature.

collapse without negative samples. Their work claims that variance component (equivalent to de-centering) is an indispensable component for preventing collapse, while we find that de-correlation itself alleviates collapse. Overall, our work helps understand various frameworks in SSL from an unified perspective, which also inspires an investigation of inter-anchor hardness-awareness Zhang et al. (2022) for further bridging the gap between CL and non-CL frameworks in SSL.

## 5 TOWARDS SIMPLIFYING THE PREDICTOR IN SIMSIAM

Based on our understanding of how SimSiam prevents collapse, we demonstrate that simple components (instead of a non-linear MLP in SimSiam) in the predictor are sufficient for preventing collapse. For example, to achieve dimensional de-correlation, a single FC layer might be sufficient because a single FC layer can realize the interaction among various dimensions. On the other hand, to achieve de-centering, a single bias layer might be sufficient because a bias vector can represent the center vector. Attaching an $l_2$-normalization layer at the end of the encoder, *i.e.* before the predictor, is found to be critical for achieving the above goal.

**Pridictor with FC layers.** To learn the dimensional correlation, an FC layer is sufficient theoretically but can be difficult to train in practice. Inspired by the property that Multiple FC layers make the training more stable even though they can be mathematically equivalent to a single FC layer (Bell-Kligler et al., 2019), we adopt two consecutive FC layers which are equivalent to removing the BN and ReLU in the original predictor.

| Method | Predictor | Top-1 (%) |
|--------|-----------|-----------|
| SimSiam | Non-linear MLP | 66.9 |
| Two FC | FC+FC+Bias | 66.7 |
| One FC | Tanh(FC) | 64.82 |
| One bias | Bias | 49.82 |

Table 5: Linear evaluation on CIFAR100.

The training can be made more stable if a Tanh layer is applied on the adopted single FC after every iteration. Table 5 shows that they achieve performance comparable to that with a non-linear MLP.

**Predictor with a bias layer.** A predictor with a single bias layer can be utilized for preventing collapse (see Table 5) and the trained bias vector is found to have a cosine similarity of 0.99 with the center vector (see Table 6). A bias in the MLP predictor also has a high cosine similarity of 0.89, suggesting that it is not a coincidence. A theoretical derivation for justifying such a high similarity as well as how this single bias layer prevents collapse are discussed in Appendix A.8.

| Bias | (1) single bias | (2) bias in MLP |
|------|-----------------|-----------------|
| Similarity | 0.99 | 0.89 |

Table 6: Similarity between *center vector* and (1) *single bias layer* ($\boldsymbol{b}_p$), (2) *the last bias layer of MLP* in the predictor.

## 6 CONCLUSION

We point out a hidden flaw in prior works for explaining the success of SimSiam and propose to decompose the representation vector and analyze the decomposed components of extra gradient. We find that its center vector gradient helps prevent collapse via the de-centering effect and its residual gradient achieves de-correlation which also alleviates collapse. Our further analysis reveals that InfoNCE achieve the two effects in a similar manner, which bridges the gap between SimSiam and InfoNCE and contributes to a unified understanding of recent progress in SSL. Towards simplifying the predictor we have also found that a single bias layer is sufficient for preventing collapse.

## ACKNOWLEDGEMENT

This work was partly supported by Institute for Information & communications Technology Planning & Evaluation (IITP) grant funded by the Korea government (MSIT) under grant No.2019-0-01396 (Development of framework for analyzing, detecting, mitigating of bias in AI model and training data), No.2021-0-01381 (Development of Causal AI through Video Understanding and Reinforcement Learning, and Its Applications to Real Environments) and No.2021-0-02068 (Artificial Intelligence Innovation Hub). During the rebuttal, multiple anonymous reviewers provide valuable advice to significantly improve the quality of this work. Thank you all.

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

# A    APPENDIX

## A.1    EXPERIMENTAL SETTINGS

**Self-supervised encoder training:** Below are the settings for self-supervised encoder training. For simplicity, we mainly use the default settings in a popular open library termed solo-learn (da Costa et al., 2021).

*Data augmentation and normalization:* We use a series of transformations including *RandomResizedCrop* with scale [0.2, 1.0], bicubic interpolation. *ColorJitter* (brightness (0.4), contrast (0.4), saturation (0.4), hue (0.1)) is randomly applied with the probability of 0.8. Random gray scale *RandomGrayscale* is applied with $p = 0.2$ Horizontal flip is applied with $p = 0.5$ The images are normalized with the mean (0.4914, 0.4822, 0.4465) and Std (0.247, 0.243, 0.261).

*Network architecture and initialization:* The backbone architecture is ResNet-18. The projection head contains three fully-connected (FC) layers followed by Batch Norm (BN) and ReLU, for which ReLU in the final FC layer is removed, *i.e.* $FC_1 + BN + ReLU + FC_2 + BN + ReLU + FC_3 + BN$. All projection FC layers have 2048 neurons for input, output as well as the hidden dimensions. The predictor head includes two FC layers as follows: $FC_1 + BN + ReLU + FC_2$. Input and output of the predictor both have the dimension of 2048, while the hidden dimension is 512. All layers of the network are by default initialized in Pytorch.

*Optimizer:* SGD optimizer is used for the encoder training. The batch size $M$ is 256 and the learning rate is linearly scaled by the formula $lr \times M/256$ with the base learning rate $lr$ set to 0.5. The schedule for learning rate adopts the cosine decay as SimSiam. Momentum 0.9 and weight decay $1.0 \times 10^{-5}$ are used for SGD. We use one GPU for each pre-training experiment. Following the practice of SimSiam, the learning rate of the predictor is fixed during the training. We use warmup training for the first 10 epochs. If not specified, by default we train the model for 1000 epochs.

**Online linear evaluation:** For the online linear revaluation, we also follow the practice in the solo-learn library (da Costa et al., 2021). The frozen features (2048 dimensions) from the training set are extracted (from the self-supervised pre-trained model) to feed into a linear classifier (1 FC layer with the input 2048 and output of 100). The test is performed on the validation set. The learning rate for the linear classifier is 0.1. Overall, we report Top-1 accuracy with the online linear evaluation in this work.

## A.2    TWO SUB-PROBLEMS IN AO OF SIMSIAM

In the sub-problem $\eta^t \leftarrow \arg\min_\eta \mathcal{L}(\theta^t, \eta)$, $\eta^t$ indicating latent representation of images at step $t$ is actually obtained through $\eta_x^t \leftarrow \mathbb{E}_{\mathcal{T}}\left[\mathcal{F}_{\theta^t}(\mathcal{T}(x))\right]$, where they in practice ignore $\mathbb{E}_{\mathcal{T}}[\cdot]$ and sample only one augmentation $\mathcal{T}'$, *i.e.* $\eta_x^t \leftarrow \mathcal{F}_{\theta^t}(\mathcal{T}'(x))$. Conceptually, Chen & He equate the role of predictor to EOA.

## A.3    EXPERIMENTAL DETAILS FOR EXPLICIT EOA IN TABLE 1

In the *Moving average* experiment, we follow the setting in SimSiam (Chen & He, 2021) without predictor. In the *Same batch* experiment, multiple augmentations, 10 augmentations for instance, are applied on the same image. With multi augmentations, we get the corresponding encoded representation, *i.e.* $z_i$, $i \in [1, 10]$. We minimize the cosine distance between the first representation $z_1$ and the average of the remaining vectors, *i.e.* $\bar{z} = \frac{1}{9}\sum_{i=2}^{10} z_i$. The gradient stop is put on the averaged vector. We also experimented with letting the gradient backward through more augmentations, however, they consistently led to collapse.

## A.4 EXPERIMENTAL SETUP AND RESULT TREND FOR TABLE 2.

**Mirror SimSiam.** Here we provide the pseudocode for Mirror SimSiam. In the Mirror SimSiam experiment which relates to Fig. 1 (c). Without taking symmetric loss into account, the pseudocode is shown in Algorithm 1. Taking symmetric loss into account, the pseudocode is shown in Algorithm 2.

---

**Algorithm 1** Pytorch-like Pseudocode: Mirror SimSiam

---

```
# f: encoder (backbone + projector)
# h: predictor

for x in loader: # load a minibatch x with n samples
    x_a, x_b = aug(x), aug(x) # augmentation
    z_a, z_b = f(x_a), f(x_b) # projections

    p_b = h(z_b.detach()) # detach z_b but still allowing gradient p_b

    L = D_cosine(z_a, p_b) # loss

    L.backward() # back-propagate
    update(f, h) # SGD update

def D_cosine(z, p): # negative cosine similarity
    z = normalize(z, dim=1) # l2-normalize
    p = normalize(p, dim=1) # l2-normalize
    return -(z*p).sum(dim=1).mean()
```

---

---

**Algorithm 2** Pytorch-like Pseudocode: Mirror SimSiam

---

```
# f: encoder (backbone + projector)
# h: predictor

for x in loader: # load a minibatch x with n samples
    x_a, x_b = aug(x), aug(x) # augmentation
    z_a, z_b = f(x_a), f(x_b) # projections

    p_b = h(z_b.detach()) # detach z_b but still allowing gradient p_b
    p_a = h(z_a.detach()) # detach z_a but still allowing gradient p_a

    L = D_cosine(z_a, p_b)/2 + D_cosine(z_b, p_a)/2 # loss

    L.backward() # back-propagate
    update(f, h) # SGD update

def D_cosine(z, p): # negative cosine similarity
    z = normalize(z, dim=1) # l2-normalize
    p = normalize(p, dim=1) # l2-normalize
    return -(z*p).sum(dim=1).mean()
```

---

**Symmetric Predictor.** To implement the SimSiam with Symmetric Predictor as in Fig. 2 (b), we can just perceive the predictor as part of the new encoder, for which the pseudocode is provided in Algorithm 3. Alternatively, we can additionally train the predictor similarly as that in SimSiam, for which the training involves two losses, one for training the predictor and another for training the new encoder (the corresponding pseudocode is provided in Algorithm 4). Moreover, for the second implementation, we also experiment with another variant that fixes the predictor while optimizing the new encoder and then train the predictor alternatingly. All of them lead to collapse with a similar trend as long as the symmetric predictor is used for training the encoder. For avoiding redundancy, in Fig. 8 we only report the result of the second implementation.

**Result trend.** The result trend of SimSiam, Naive Siamese, Mirror SimSiam, Symmetric Predictor are shown in Fig. 8. We observe that all architectures lead to collapse except for SimSiam. Mirroe SimSiam was stopped in the middle because a NaN value was returned from the loss.

## A.5 EXPERIMENTAL DETAILS FOR INVERSE PREDICTOR.

In the inverse predictor experiment which relates to Fig. 2 (c), we introduce a new predictor which has the same structure as that of the original predictor. The training loss consists of 3 parts: predictor training loss, inverse predictor training and new encoder (old encoder+predictor) training. The new

**Algorithm 3** Pytorch-like Pseudocode: Symmetric Predictor

```
# f: encoder (backbone + projector)
# h: predictor

for x in loader: # load a minibatch x with n samples
    x_a, x_b = aug(x), aug(x) # augmentation
    z_a, z_b = f(x_a), f(x_b) # projections
    p_a, p_b = h(z_a), h(z_b) # predictions

    L = D(p_a, p_b)/2 + D(p_b, p_a)/2 # loss

    L.backward() # back-propagate
    update(f, h) # SGD update

def D(p, z): # negative cosine similarity
    z = z.detach() # stop gradient
    p = normalize(p, dim=1) # l2-normalize
    z = normalize(z, dim=1) # l2-normalize
    return -(p*z).sum(dim=1).mean()
```

**Algorithm 4** Pytorch-like Pseudocode: Symmetric Predictor (with additional training on predictor)

```
# f: encoder (backbone + projector)
# h: predictor

for x in loader: # load a minibatch x with n samples
    x_a, x_b = aug(x), aug(x) # augmentation
    z_a, z_b = f(x_a), f(x_b) # projections
    p_a, p_b = h(z_a), h(z_b) # predictions

    d_p_a, d_p_b = h(z_a.detach()), h(z_b.detach()) # detached predictor output

    # predictor training loss
    L_pred = D(d_p_a, z_b)/2 + D(d_p_b, z_a)/2

    # encoder training loss
    L_enc = D(p_a, d_p_b)/2 + D(p_b, d_p_a)/2

    L = L_pred + L_enc

    L.backward() # back-propagate
    update(f, h) # SGD update

def D(p, z): # negative cosine similarity with detach on z
    z = z.detach() # stop gradient
    p = normalize(p, dim=1) # l2-normalize
    z = normalize(z, dim=1) # l2-normalize
    return -(p*z).sum(dim=1).mean()
```

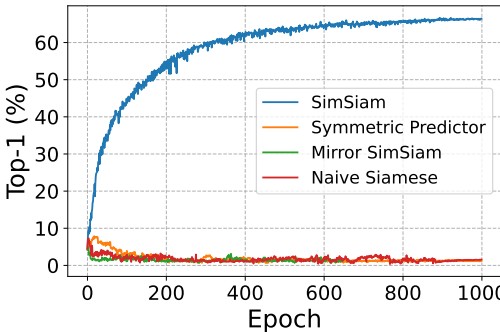

Figure 8: Result trend of Naive Siamese, Mirror SimSiam, Symmetric Predictor.

encoder $F$ consists of the old encoder $f$ + predictor $h$. The practice of gradient stop needs to be considered in the implementation. We provide the pseudocode in Algorithm 5.

---

**Algorithm 5** Pytorch-like Pseudocode: Trainable Inverse Predictor

---

```
# f: encoder (backbone + projector)
# h: predictor
# h_inv: inverse predictor

for x in loader: # load a minibatch x with n samples
    x_a, x_b = aug(x), aug(x) # augmentation
    z_a, z_b = f(x_a), f(x_b) # projections
    p_a, p_b = h(z_a), h(z_b) # predictions

    d_p_a, d_p_b = h(z_a.detach()), h(z_b.detach()) # detached predictor output
    # predictor training loss
    L_pred = D(d_p_a, z_b)/2 + D(d_p_b, z_a)/2 # to train h

    inv_p_a, inv_p_b = h_inv(p_a.detach()), h_inv(p_b.detach()) # to train h_inv
    # inverse predictor training loss
    L_inv_pred = D(inv_p_a, z_a)/2 + D(inv_p_b, z_b)/2

    # encoder training loss
    L_enc = D(p_a, h_inv(p_b))/2 + D(p_b, h_inv(p_a))

    L = L_pred + L_inv_pred + L_enc

    L.backward() # back-propagate
    update(f, h, h_inv) # SGD update

def D(p, z): # negative cosine similarity with detach on z
    z = z.detach() # stop gradient
    p = normalize(p, dim=1) # l2-normalize
    z = normalize(z, dim=1) # l2-normalize
    return -(p*z).sum(dim=1).mean()
```

---

## A.6 Regularization loss

Following Zbontar et al. (2021), we compute covariance regularization loss of encoder output along the mini-batch. The pseudocode for de-correlation loss calculation is put in Algorithm 6.

---

**Algorithm 6** Pytorch-like Pseudocode: De-correlation loss

---

```
# Z_a: representation vector
# N: batch size
# D: the number of dimension for representation vector

Z_a = Z_a - Z_a.mean(dim=0)

cov = Z_a.T @ Z_a / (N-1)
diag = torch.eye(D)

loss = cov[~diag.bool()].pow_(2).sum() / D
```

---

## A.7 Gradient derivation and temperature analysis for InfoNCE

With · indicating the cosine similarity between vectors, the InfoNCE loss can be expressed as

$$
\begin{aligned}
\mathcal{L}_{InfoNCE} &= -\log \frac{\exp(\boldsymbol{Z}_a \cdot \boldsymbol{Z}_b / \tau)}{\exp(\boldsymbol{Z}_a \cdot \boldsymbol{Z}_b / \tau) + \sum_{i=1}^{N} \exp(\boldsymbol{Z}_a \cdot \boldsymbol{Z}_i / \tau)} \\
&= -\log \frac{\exp(\boldsymbol{Z}_a \cdot \boldsymbol{Z}_b / \tau)}{\sum_{i=0}^{N} \exp(\boldsymbol{Z}_a \cdot \boldsymbol{Z}_i / \tau)},
\end{aligned}
\tag{5}
$$

where $N$ indicates the number of negative samples and $\boldsymbol{Z}_0 = \boldsymbol{Z}_b$ for simplifying the notation. By treating $\boldsymbol{Z}_a \cdot \boldsymbol{Z}_i$ as the logit in a normal CE loss, we have the corresponding probability for each negative sample as $\lambda_i = \frac{\exp(\boldsymbol{Z}_a \cdot \boldsymbol{Z}_i / \tau)}{\sum_{i=0}^{N} \exp(\boldsymbol{Z}_a \cdot \boldsymbol{Z}_i / \tau)}$, where $i = 0, 1, 2, ..., N$ and we have $\sum_{i=0}^{N} \lambda_i = 1$.

The negative gradient of the InfoNCE on the representation $\boldsymbol{Z}_a$ is shown as

$$
\begin{aligned}
-\frac{\partial \mathcal{L}_{InfoNCE}}{\partial \boldsymbol{Z}_a} &= \frac{1}{\tau}(1-\lambda_0)\boldsymbol{Z}_b - \frac{1}{\tau}\sum_{i=1}^{N}\lambda_i\boldsymbol{Z}_i \\
&= \frac{1}{\tau}(\boldsymbol{Z}_b - \sum_{i=0}^{N}\lambda_i\boldsymbol{Z}_i) \\
&= \frac{1}{\tau}(\boldsymbol{Z}_b - \sum_{i=0}^{N}\lambda_i(\boldsymbol{o}_z + \boldsymbol{r}_i)) \\
&= \frac{1}{\tau}(\boldsymbol{Z}_b + (-\boldsymbol{o}_z - \sum_{i=0}^{N}\lambda_i\boldsymbol{r}_i) \\
&\propto \boldsymbol{Z}_b + (-\boldsymbol{o}_z - \sum_{i=0}^{N}\lambda_i\boldsymbol{r}_i)
\end{aligned}
\tag{6}
$$

where $\frac{1}{\tau}$ can be adjusted through learning rate and is omitted for simple discussion. With $\boldsymbol{Z}_b$ as the basic gradient, $\boldsymbol{G}_e = -\boldsymbol{o}_z - \sum_{i=0}^{N}\lambda_i\boldsymbol{r}_i$, for which $\boldsymbol{o}_e = -\boldsymbol{o}_z$ and $\boldsymbol{r}_e = -\sum_{i=0}^{N}\lambda_i\boldsymbol{r}_i$.

When the temperature is set to a large value, $\lambda_i = \frac{\exp(\boldsymbol{Z}_a\cdot\boldsymbol{Z}_i/\tau)}{\sum_{i=0}^{N}\exp(\boldsymbol{Z}_a\cdot\boldsymbol{Z}_i/\tau)}$, approaches $\frac{1}{N+1}$, indicated by a high entropy value (see Fig. 7). InfoNCE will degenerate to a simple contrastive loss, *i.e.* $\mathcal{L}_{simple} = -\boldsymbol{Z}_a \cdot \boldsymbol{Z}_b + \frac{1}{N+1}\sum_{i=0}^{N}\boldsymbol{Z}_a \cdot \boldsymbol{Z}_i$ , which repulses every negative sample with an equal force. In contrast, a relative smaller temperature will give more relative weight, *i.e.* larger $\lambda$, to negative samples that are more similar to the anchor ($\boldsymbol{Z}_a$).

The influence of the temperature on the covariance and accuracy is shown in Fig. 7 (b) and (c). We observe that a higher temperature tends to decrease the effect of de-correlation, indicated by a higher covariance value, which also leads to a performance drop. This verifies our hypothesis regarding on how $\boldsymbol{r}_e$ in InfoNCE achieves de-correlation because a large temperature causes more balanced weights $\lambda_i$, which is found to alleviate the effect of de-correlation. For the setup, we note that the encoder is trained for 200 epochs with the default setting in Solo-learn for the SimCLR framework.

### A.8 THEORETICAL DERIVATION FOR A SINGLE BIAS LAYER

With the cosine similarity loss defined as Eq 7 Eq 8:

$$
cossim(a,b) = \frac{a\cdot b}{\sqrt{a^2\cdot b^2}},
\tag{7}
$$

for which the derived gradient on the vector $a$ is shown as

$$
\frac{\partial}{\partial a}cossim(a,b) = \frac{b_1}{|a|\cdot|b|} - cossim(a,b)\cdot\frac{a_1}{|a|^2}.
\tag{8}
$$

The above equation is used as a prior for our following derivations. As indicated in the main manuscript, the encoder output $z_a$ is $l_2$-normalized before feeding into the predictor, thus $\boldsymbol{p}_a = \boldsymbol{Z}_a + \boldsymbol{b}_p$, $\boldsymbol{b}_p$ denotes the bias layer in the predictor. The cosine similarity loss (ignoring the symmetry for simplicity) is shown as

$$
\begin{aligned}
\mathcal{L}_{cosine} &= -\boldsymbol{P}_a\cdot\boldsymbol{Z}_b \\
&= -\frac{\boldsymbol{p}_a}{||\boldsymbol{p}_a||}\cdot\frac{\boldsymbol{z}_b}{||\boldsymbol{z}_b||}
\end{aligned}
\tag{9}
$$

The gradient on $\boldsymbol{p}_a$ is derived as

$$
\begin{aligned}
-\frac{\partial \mathcal{L}_{cosine}}{\partial \boldsymbol{p}_a} &= \frac{\boldsymbol{z}_b}{\|\boldsymbol{z}_b\| \cdot \|\boldsymbol{p}_a\|} - cossim(\boldsymbol{Z}_a, \boldsymbol{Z}_b) \cdot \frac{\boldsymbol{p}_a}{\|\boldsymbol{p}_a\|^2} \\
&= \frac{1}{\|\boldsymbol{p}_a\|} \left( \frac{\boldsymbol{z}_b}{\|\boldsymbol{z}_b\|} - cossim(\boldsymbol{Z}_a, \boldsymbol{Z}_b) \cdot \boldsymbol{P}_a \right) \\
&= \frac{1}{\|\boldsymbol{p}_a\|} \left( \boldsymbol{Z}_b - cossim(\boldsymbol{Z}_a, \boldsymbol{Z}_b) \cdot \frac{\boldsymbol{Z}_a + \boldsymbol{b}_p}{\|\boldsymbol{p}_a\|} \right) \\
&= \frac{1}{\|\boldsymbol{p}_a\|} \left( (\boldsymbol{o}_z + \boldsymbol{r}_b) - \frac{cossim(\boldsymbol{Z}_a, \boldsymbol{Z}_b)}{\|\boldsymbol{p}_a\|} \cdot (\boldsymbol{o}_z + \boldsymbol{r}_a + \boldsymbol{b}_p) \right) \\
&= \frac{1}{\|\boldsymbol{p}_a\|} \left( (\boldsymbol{o}_z + \boldsymbol{r}_b) - m \cdot (\boldsymbol{o}_z + \boldsymbol{r}_a + \boldsymbol{b}_p) \right) \\
&= \frac{1}{\|\boldsymbol{p}_a\|} \left( (1 - m)\boldsymbol{o}_z - m\boldsymbol{b}_p + \boldsymbol{r}_b - m \cdot \boldsymbol{r}_a \right),
\end{aligned}
\tag{10}
$$

where $m = \frac{cossim(\boldsymbol{Z}_a, \boldsymbol{Z}_b)}{\|\boldsymbol{p}_a\|}$.

Given that $\boldsymbol{p}_a = \boldsymbol{Z}_a + \boldsymbol{b}_p$, the negative gradient on $\boldsymbol{b}_p$ is the same as that on $\boldsymbol{p}_a$ as

$$
\begin{aligned}
-\frac{\partial \mathcal{L}_{cosine}}{\partial \boldsymbol{b}_p} &= -\frac{\partial \mathcal{L}_{cosine}}{\partial \boldsymbol{p}_a} \\
&= \frac{1}{\|\boldsymbol{p}_a\|} \left( (1 - m)\boldsymbol{o}_z - m\boldsymbol{b}_p + \boldsymbol{r}_b - m \cdot \boldsymbol{r}_a \right).
\end{aligned}
\tag{11}
$$

We assume that the training is stable and the bias layer converges to a certain value when $-\frac{\partial cossim(\boldsymbol{Z}_a, \boldsymbol{Z}_b)}{\partial \boldsymbol{b}_p} = 0$. Thus, the converged $\boldsymbol{b}_p$ satisfies the following constraint:

$$
\begin{aligned}
\frac{1}{\|\boldsymbol{p}_a\|} \left( (1 - m)\boldsymbol{o}_z - m\boldsymbol{b}_p + \boldsymbol{r}_b - m\boldsymbol{r}_a) \right) &= 0 \\
\boldsymbol{b}_p &= \frac{1 - m}{m}\boldsymbol{o}_z + \frac{1}{m}\boldsymbol{r}_b - \boldsymbol{r}_a.
\end{aligned}
\tag{12}
$$

With a batch of samples, the average of $\frac{1}{m}\boldsymbol{r}_b$ and $\boldsymbol{r}_a$ is expected to be close to 0 by the definition of residual vector. Thus, the bias layer vector is expected to converge to:

$$
\boldsymbol{b}_p = \frac{1 - m}{m}\boldsymbol{o}_z.
\tag{13}
$$

**Rational behind the high similarity between $\boldsymbol{b}_p$ and $\boldsymbol{o}_z$.** The above theoretical derivation shows that the parameters in the bias layer are excepted to converge to a vector $\frac{1-m}{m}\boldsymbol{o}_z$. This theoretical derivation justifies why the empirically observed cosine similarity between $\boldsymbol{b}_p$ and $\boldsymbol{o}_z$ is as high as 0.99. Ideally, it should be 1, however, such a small deviation is expected with the training dynamics taken into account.

**Rational behind how a single bias layer prevents collapse.** Given that $\boldsymbol{p}_a = \boldsymbol{Z}_a + \boldsymbol{b}_p$, the negative gradient on $\boldsymbol{Z}_a$ is shown as

$$
\begin{aligned}
-\frac{\partial \mathcal{L}_{cosine}}{\partial \boldsymbol{Z}_a} &= -\frac{\partial \mathcal{L}_{cosine}}{\partial \boldsymbol{p}_a} \\
&= \frac{1}{\|\boldsymbol{p}_a\|} \left( \boldsymbol{Z}_b - cossim(\boldsymbol{Z}_a, \boldsymbol{Z}_b) \cdot \frac{\boldsymbol{Z}_a + \boldsymbol{b}_p}{\|\boldsymbol{p}_a\|} \right) \\
&= \frac{1}{\|\boldsymbol{p}_a\|} \boldsymbol{Z}_b - \frac{cossim(\boldsymbol{Z}_a, \boldsymbol{Z}_b)}{\|\boldsymbol{p}_a\|^2} \boldsymbol{Z}_a - \frac{cossim(\boldsymbol{Z}_a, \boldsymbol{Z}_b)}{\|\boldsymbol{p}_a\|^2} \boldsymbol{b}_p.
\end{aligned}
\tag{14}
$$

Here, we highlight that since the loss $-\boldsymbol{Z}_a \cdot \boldsymbol{Z}_a = -1$ is a constant having zero gradients on the encoder, $-\frac{cossim(\boldsymbol{Z}_a, \boldsymbol{Z}_b)}{\|\boldsymbol{p}_a\|^2} \boldsymbol{Z}_a$ can be seen as a *dummy* term. Considering Eq 13 and $m = \frac{cossim(\boldsymbol{Z}_a, \boldsymbol{Z}_b)}{\|\boldsymbol{p}_a\|}$,

we have $b = (\frac{\|\boldsymbol{p}_a\|}{cossim(\boldsymbol{Z}_a, \boldsymbol{Z}_b)} - 1)\boldsymbol{o}_z$. The above equation is equivalent to

$$
\begin{aligned}
-\frac{\partial \mathcal{L}_{cosine}}{\partial \boldsymbol{Z}_a} &= \frac{1}{\|\boldsymbol{p}_a\|}\boldsymbol{Z}_b - \frac{cossim(\boldsymbol{Z}_a, \boldsymbol{Z}_b)}{\|\boldsymbol{p}_a\|^2}\boldsymbol{b}_p \\
&= \frac{1}{\|\boldsymbol{p}_a\|}\boldsymbol{Z}_b - \frac{cossim(\boldsymbol{Z}_a, \boldsymbol{Z}_b)}{\|\boldsymbol{p}_a\|^2}(\frac{\|\boldsymbol{p}_a\|}{cossim(\boldsymbol{Z}_a, \boldsymbol{Z}_b)} - 1)\boldsymbol{o}_z \\
&= \frac{1}{\|\boldsymbol{p}_a\|}\boldsymbol{Z}_b - \frac{1}{\|\boldsymbol{p}_a\|}(1 - \frac{cossim(\boldsymbol{Z}_a, \boldsymbol{Z}_b)}{\|\boldsymbol{p}_a\|})\boldsymbol{o}_z \\
&\propto \boldsymbol{Z}_b - (1 - \frac{cossim(\boldsymbol{Z}_a, \boldsymbol{Z}_b)}{\|\boldsymbol{p}_a\|})\boldsymbol{o}_z.
\end{aligned}
\tag{15}
$$

With $\boldsymbol{Z}_b$ as the basic gradient, the extra gradient component $\boldsymbol{G}_e = -(1 - \frac{cossim(\boldsymbol{Z}_a, \boldsymbol{Z}_b)}{\|\boldsymbol{p}_a\|})\boldsymbol{o}_z$. Given that $\boldsymbol{p}_a = \boldsymbol{Z}_a + \boldsymbol{b}_p$ and $\|\boldsymbol{Z}_a\| = 1$, thus $\|\boldsymbol{p}_a\| < 1$ only when $\boldsymbol{Z}_a$ is negatively correlated with $\boldsymbol{b}_p$. In practice, however, $\boldsymbol{Z}_a$ and $\boldsymbol{b}_p$ are often positively correlated to some extent due to their shared center vector component. In other words, $\|\boldsymbol{p}_a\| > 1$. Moreover, $cossim(\boldsymbol{Z}_a, \boldsymbol{Z}_b)$ is smaller than 1, thus $-(1 - \frac{cossim(\boldsymbol{Z}_a, \boldsymbol{Z}_b)}{\|\boldsymbol{p}_a\|}) < 0$, suggesting $\boldsymbol{G}_e$ consists of negative $\boldsymbol{o}_z$ with the effect of de-centerization. This above derivation justifies the rationale why a single bias layer can help alleviate collapse.

# B   DISCUSSION: DOES BN HELP AVOID COLLAPSE?

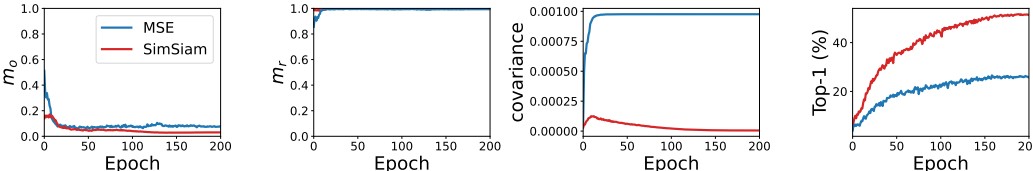

Figure 9: BN with MSE helps prevent collapse without predictor or stop gradient. Its performance, however, is inferior to the cosine loss-based SimSiam (with predictor and stop gradient).

To our knowledge, our work is the first to revisit and refute the explanatory claims in (Chen & He, 2021). Several works, however, have attempted to demystify the success of BYOL (Grill et al., 2020), a close variant of SimSiam. The success has been ascribed to BN in (Fetterman & Albrecht, 2020), however, (Richemond et al., 2020) refutes their claim. Since the role of intermediate BNs is ascribed to stabilize training (Richemond et al., 2020; Chen & He, 2021), we only discuss the final BN in the SimSiam encoder. Note that with our Conjecture1, the final BN that removes the mean of representation vector is supposed to have de-centering effect. BY default SimSiam has such a BN at the end of its encoder, however, it still collapses with the predictor and stop gradient. Why would such a BN not prevent collapse in this case? Interestingly, we observe that such BN can help alleviate collapse with a simple MSE loss (see Fig. 9), however, its performance is is inferior to the cosine loss-based SimSiam (with predictor and stop gradient) due to the lack of the de-correlation effect in SimSiam. Note that the cosine loss is in essence equivalent to a MSE loss on the $l_2$-normalized vectors. This phenomenon can be interpreted as that the $l_2$-normalization causes another mean after the BN removes it. Thus, with such $l_2$-normalization in the MSE loss, *i.e.* adopting the default cosine loss, it is important to remove the $\boldsymbol{o}_e$ from the optimization target. The results with the loss of $-\boldsymbol{Z}_a \cdot \text{sg}(\boldsymbol{Z}_b + \boldsymbol{o}_e)$ in Table 3 show that this indeed prevents collapse and verifies the above interpretation.

