# OpenReview forum: "How Does SimSiam Avoid Collapse Without Negative Samples? A Unified Understanding with Self-supervised Contrastive Learning"
_ICLR.cc/2022/Conference — ICLR 2022 Poster_

### Official Review · Reviewer_S8KB · 2021-10-18

**Correctness:** 4
**Technical Novelty And Significance:** 4
**Empirical Novelty And Significance:** 4
**Recommendation:** 8
**Confidence:** 3

**Main Review:**

This paper is generally well-written and well-structured. It is important and inspiring to explore the reason why SimSiam can avoid collapse. The authors provide a convincing explanation and reach a unified conclusion for the recent progress in SSL, which is very insightful. The theoretical analysis and experimental results are solid.

**Summary Of The Paper:**

This method aims to explore why a minimalist simple Siamese (SimSiam) method can avoid collapse. After refuting their claims, the authors introduce vector decomposition for analyzing the collapse based on the gradient analysis of l2 normalized vector, yielding a unified perspective on how negative samples and SimSiam predictor alleviate collapse.

**Summary Of The Review:**

Based on the abovementioned strength of this paper and the interesting conclusion, I recommend to accept this paper.

---

> ### Author Response · Authors · 2021-11-19
> **First reply**
>
> Thank you very much for your appreciation for this paper.
>
>  We polished the paper in the revised version.
>
> We hope you enjoy the revised version!

---

> ### Author Response · Authors · 2021-11-23
> **Second reply**
>
> > #### Summary of The Review: Based on the above mentioned strength of this paper and the interesting conclusion, I recommend to accept this paper.
>
> Thanks for encouraging remarks. We hope you enjoy our revised final paper.

---

### Official Review · Reviewer_D1d5 · 2021-10-29

**Correctness:** 3
**Technical Novelty And Significance:** 3
**Empirical Novelty And Significance:** 2
**Recommendation:** 6
**Confidence:** 3

**Main Review:**

#### Strengths:
+ Recent successful approaches to SSL indicate that stable learning can be achieved without explicit incorporation of repulsive, negative image relations. Indeed, the underlying reasons are theoretically poorly understood and typically only an inuitive understanding presented. Hence, given the importance of contrastive learnig today, research in this direction is important.
+ The analysis in section 2 seems sound and raises questions about the original insights of the SimSiam work regarding representation collapse.

#### Weaknesses:
- The presentation and outline of arguments, as well as empirical discussions especially in section 3.3, 3.4 and 3.5 are sometimes hard to follow.
- The need of ‘de-centeralization’ (i.e. pushing samples apart in the embedding space) in gradient signals of SSL for stable learning seems straight-forward and inuitively trivial and does not seem novel.
- The paper only empirically shows that some kind of repulsion (i.e. de-centralization) is implicitly happening in the SimSiam framework. However, no theoretical and clear explanation of why and how is presented.  The presented conjectures are insufficiently back-uped and proven beyond the intuition of implicit repulsion. The paper e.g. states “Since the original predictor involves a nonlinear MLP, it is hard to understand how the predictor actually achieves it” (Sec. 4). Thus, no real, novel insights about the learning mechanisms of SSL without negatives are provided which are required for a more complete understanding of the addressed problem.
- The conceptual explanation of de-correlation in Info-NCE is hard is fuzzy and does not sound convincing. A more solid derivation and formulation of the hypothesis is needed.

#### Questions:
- In the paragraph ‘Predictor with stop-gradient is asymmetric’ the paper states that the model setup shown in Fig. 2 (a) acutally results in successful, stable learning (“Similarly, […] Fig2 (a) […] leads to success, [...]”) It seems no prediction head is used in this case. Does the stop-gradient operation itself already prevent representation collapse?

**Summary Of The Paper:**

The paper analyzes how the self-supervised learning (SSL) approach SimSiam avoids collapsed representations without explicit formulation of repulsive sample relations. To this end, first flaws in the original reasoning of the SimSiam paper are revealed. Next, based on center-residual vector decomposition, the role of the prediction head for preventing representation collapse in SimSiam is analyzed. Results indicate the importance of de-centralization and de-correlation as driving concepts for stable SSL.

**Summary Of The Review:**

While the addressed phenomenon of preventing collapsing representations without the usage of negative samples is interesting, the presented analyis provides insufficient novel insights about how collapse is prevented. Although some novel framework based on vector decomposition is presented, the reasoning is often fuzzy, not convincing and lacks backup by clear theoretical derivations. Thus, I vote to reject this submission in its present form.

---

> ### Author Response · Authors · 2021-11-19
> **First reply**
>
> Many thanks for the great comments! Please find our replies below.
> ### Weakness 1: The presentation and outline of arguments, as well as empirical discussions especially in section 3.3, 3.4 and 3.5 are sometimes hard to follow.
> The reply to all reviewers summarizes the changes made in the revised paper. For more detail, please check our revised submission. If anything remains unclear, please kindly let us know.
>
> ### Weakness 2: The need of ‘de-centeralization’ (i.e. pushing samples apart in the embedding space) in gradient signals of SSL for stable learning seems straight-forward and inuitively trivial and does not seem novel.
>
> We agree that de-centeralization for avoiding collapse seems straightforward. However, we argue that they are not necessarily trivial, especially in the context of SimSiam.
>
>
> First, we are the first to propose to decompose the representation into the center vector and residual vector. Prior works have mainly ambiguously attributed the collapse to a repulsive part. We explicitly show that this repulsive force is the center vector instead of the residual vector. Leaving it ambiguous might mislead readers to believe that the residual vector is what prevents collapse because the residual vector captures it as a negative sample.
>
> Second, we argue that our finding is not trivial in the context of SimSiam. Otherwise, the authors of SimSiam would not have attempted the explanation from the perspective of EOA which is refuted by our work. Our work is the first to show that how SimSiam avoids collapse can be understood from the **straightforward** de-centerization. Moreover, we are the first to show that de-correlation also helps prevent collapse. In other words, de-centerization is not the only effect for avoiding collase, which deserves the attention of the community.
>
> ### Weakness 3: The paper only empirically shows that some kind of repulsion (i.e. de-centralization) is implicitly happening in the SimSiam framework. However, no theoretical and clear explanation of why and how is presented. The presented conjectures are insufficiently backuped and proven beyond the intuition of implicit repulsion. The paper e.g. states “Since the original predictor involves a nonlinear MLP, it is hard to understand how the predictor actually achieves it” (Sec. 4). Thus, no real, novel insights about the learning mechanisms of SSL without negatives are provided which are required for a more complete understanding of the addressed problem.
>
> First of all, we find that both de-centerization and de-correlation help prevent collapse. Both of them are proposed as conjecture and then corroborated by empirical results. We show that they can be used to explain how negative samples in InfoNCE and negative-free SimSiam avoid collapse in a unified manner.
>
> Second, with the focus on SimSiam from the perspective of de-centerization, our work reveals interesting findings. SimSiam shows that it can avoid collapse without negative samples. However, we show that they still implicittly use negative samples (through $o_z$). With a single bias layer, we perform a theoretical derivation of what the bias vector converges to. We find that the bias vector roughly converges to $k o_z$ where k is a certain (dynamically changing) scalar vector in Appendix A.8. In other words, $o_z$ is implicitly used. We empirically verify our derivation by showing the cosine similarity between the bias vector and center vector is surprisingly high 0.99 (see "Table 6" of the revised paper). Note that our simple yet explainable predictor design is based on our understanding of the original SimSiam predictor with non-linear MLP. Indeed, based on the original non-linear MLP, it is non-tractable to provide a theoretical and clear explanation. However, a single bias layer facilitates this (see Appendix A.8). Finally, we note that if a single bias layer can achieve this goal, it is fully expected that a non-linear MLP can achieve the same goal.
>
> ### Weakness 4:  The conceptual explanation of de-correlation in Info-NCE is hard is fuzzy and does not sound convincing. A more solid derivation and formulation of the hypothesis is needed.
>
> We have revised the relevant content, mainly in two ways. First, we remove those fuzzy and wordy sentences and summarize them as more simple intuitive explanations. Second, as suggested, we have added a solid derivation, based on which we formulate a hypothesis that the de-correlation of InfoNCE comes from that the gradient put different weights on the negative samples based on how similar it is to the anchor sample. This hypothesis is corroborated by the analysis of temperature in Appendix A.7.
>
> ### Questions: “Similarly, […] Fig2 (a) […] leads to success, [...]”. Does the stop-gradient operation itself already prevent representation collapse?
>
> Stop-gradient itself does not prevent representation collapse. Fig. 1\(c\) and Fig. 2(a) both lead to **failure**. We apologize for the confusion caused by such a typo.

---

> ### Author Response · Authors · 2021-11-23
> **Second reply**
>
> > #### Summary of The Review: "While the addressed phenomenon of preventing collapsing representations without the usage of negative samples is interesting, the presented analyis provides insufficient novel insights about how collapse is prevented. Although some novel framework based on vector decomposition is presented, the reasoning is often fuzzy, not convincing and lacks backup by clear theoretical derivations. Thus, I vote to reject this submission in its present form.""
>
> Thanks for recognizing that we are addressing an interesting problem. Here, we summarize our novelty items.
> - We are the first to refute the claim in the original SimSiam via pointing out their reasoning flaw as well experimental results, such as explicit EOA.
> - We are the first to propose vector decomposition for analyzing the collapse in SSL. We identify that the center vector component has the effect of de-centering to prevent collapse, while its residual vector component achieves the effect of de-correlation which also helps alleviate collapse. Regarding the de-correlation to prevent collapse, we are the first to report this phenomenon and analyze its reason.
> - Based on our gradient decomposition framework, we reveal that InfoNCE with negative samples also achieves the de-centering and de-correlation effect. This yields a unified perspective on SimSiam and InfoNCE and bridges their gap with those SSL frameworks with explicit de-centering and de-correlation.
> - Our understanding of how SimSiam prevents collapse motivates us to simplify the predictor in SimSiam.  Interestingly, we have found that a single bias layer can be sufficient for preventing collapse, which has been justified by a theoretical derivation in Appendix A.8
>
> > #### Fuzzy reasoning for de-correlation in InfoNCE and lack of theoretical derivation.
> - For the InfoNCE, we have added solid derivation for the gradient of InfoNCE in Appendix A.7 and proposed a hypothesis that the de-correlation of InfoNCE arises from the gradient putting different weights on negative samples based on their similarity to the anchor. We verify this hypothesis through the results in Fig. 7.
> - Our analysis of SimSiam in Sec 3.4 is based on the original SimSiam with a non-linear MLP predictor. The new understanding motivates us to simplify the predictor and we have found that a single bias layer can be sufficient for preventing collapse. it is very challenging to perform a theoretical derivation on a non-linear MLP predictor, however, how a single bias layer prevents collapse is justified by a theoretical derivation in Appendix A.8. We highlight that it is well expected that a non-linear MLP can prevent collapse if a single bias layer is sufficient.
>
> **Could you take the time to check our reply as well as the revised final paper? Thank you very much.**

---

> ### Comment · Reviewer_D1d5 · 2021-11-28
> **Good Rebuttal**
>
> I acknowledge and thank the authors for the effort put into the rebuttal and update of the submitted manuscript. While I am still not fully convinced by the theoretical analysis based on center and residual vector decomposition, I still increase my rating to 6 (marginally above the acceptance threshold).

---

> > ### Author Response · Authors · 2021-11-28
> > **Thanks for increasing the rating score**
> >
> > We thank Reviewer D1d5 for checking our reply as well as our revised manuscript. We have spent tremendous effort on addressing the concerns and updating the submitted manuscript. We thank Reviewer D1d5 for recognizing this effort as a good rebuttal and increasing the rating. It is very encouraging!

---

### Official Review · Reviewer_d9ds · 2021-11-05

**Correctness:** 2
**Technical Novelty And Significance:** 2
**Empirical Novelty And Significance:** 2
**Recommendation:** 6
**Confidence:** 3

**Main Review:**

+ It's interesting to see a framework to unified understand SSLs such as SimSiam, MoCo, SimCLR, etc.
+ The hidden flaw in AO of SimSiam seems to be correct, which is interesting.
--------------------
- The paper lacks experimental results/details to demonstrate their statements.

a. In subsec. "Symmetric Predictor does not prevent collapse", the authors state "The results in Fig. 3 (b) show that it still leads to collapse" which is related to symmetric predictors in SimSiam. However, Fig. 3 (b) is actually about the basic SimSiam and SimSiam + Inverse predictor.

b. in subsec. "Predictor with stop gradient is asymmetric", the authors "the SimSiam avoids collapse by excluding Mirror SimSiam (Fig1 (c)) which has a loss (mirror-like Eq 2) as shown as eq. 2". There is no experimental evidences to show if the mirror SimSiam will lead to collapse. If experimentally mirror SimSiam works, the statement does not hold.

c. How did you design the inverse predictor? Can we just see it as the new predictor while the previous predictor is included in the projector part?

d. in subsec. "Predictor vs. inverse predictor", "we interpret Fig2 (b) differs from Fig1 (a) as changing the optimized target from p to z, i.e. h
−1(p), suggesting processingthe optimized target with h−1 helps prevent collapse." Again, no experimental evidence.

e. in subsec. "Trainable h−1 and its implication on EOA", "we optimize h−1 by optimizing the pa approaching z∗b while simultaneously optimizing z∗b to zb via cosine loss, where z∗b is the h−1 output. The results proves that the model with h−1 (Fig2 (c)) is equivalent to
SimSiam since it achieves comparable performance as the original SimSiam that directly optimizes pa approaching zb.". Where are the "results"?

- The paper might have flaws in their proposed math and sometimes it's hard to follow 3.1, 3.3, and 3.4

In subsec "Competition between o and r.", why "mo and mr is expected to be opposite of each other". The denominators of the two terms are both ||z||. So only ||o_z|| and ||r_a|| define the values. However, r_a = Z_a − o_z. Thus r_a and o_z can have the same norms but with different directions? Why they need to be on the opposite directions?




**Summary Of The Paper:**

This paper proposes a framework to understand why SimSiam avoid collapse without negative samples? It provides a hidden flaw of the Alternating Optimization for explain why SimSiam works. And the authors claim that  the center vector gradient has the de-centering effect and the residual gradient vector has the de-correlation effect.

**Summary Of The Review:**

I think the paper needs to improve descriptions about their framework and provide more experimental evidences.

---

> ### Author Response · Authors · 2021-11-19
> **First reply**
>
> Many thanks for the great comments! Please find our replies below.
>
> ## The paper lacks experimental results/details to demonstrate their statements.
>
> Sorry for the missing experimental results and unclear details. We have added the missing experimental results as well as a detailed setup in the revised paper.
>
> ### a. In subsec. "Symmetric Predictor does not prevent collapse", the authors state "The results in Fig. 3 (b) show that it still leads to collapse" which is related to symmetric predictors in SimSiam. However, Fig. 3 (b) is actually about the basic SimSiam and SimSiam + Inverse predictor.
>
> We provide the result of symmetric predictors in Table 2 of the revised paper. We also provide the implementation pseudocode for the result of symmetric predictor in Appendix A.4.
>
>
>
> ### b. in subsec. "Predictor with stop gradient is asymmetric", the authors "the SimSiam avoids collapse by excluding Mirror SimSiam (Fig1 \(c\)) which has a loss (mirror-like Eq 2) as shown as eq. 2". There is no experimental evidences to show if the mirror SimSiam will lead to collapse. If experimentally mirror SimSiam works, the statement does not hold.
>
> In the revised paper, we add the experimental results of Mirror SimSiam in the row of Mirror Simsiam in Table 2, which shows the Mirror SimSiam leads to collapse. The detailed trend, as well as experimental setup, are given in Appendix A.4 with Pseudocode.
>
>
>
> ### c. How did you design the inverse predictor? Can we just see it as the new predictor while the previous predictor is included in the projector part?
> The inverse predictor has the same architecture as that of original predictor. Yes, we can just see it as the new predictor while the previous predictor is included in the projector part. For more details, see appendix A.5.
>
>
>
>
>
> ### d. in subsec. "Predictor vs. inverse predictor", "we interpret Fig2 (b) differs from Fig1 (a) as changing the optimized target from p to z, i.e. $h^{-1}(p)$, suggesting processing the optimized target with $h^{-1}$ helps prevent collapse." Again, no experimental evidence.
>
> Fig. 1(a) refers to the original SimSiam and Fig. 2(b) symmetric predictor. The results are summarized in Table 2 of the revised paper.
>
>
>
>
> ### e. in subsec. "Trainable  $h^{-1}$ and its implication on EOA", "we optimize  $h^{-1}$ by optimizing the $p_a$ approaching $z_b$ while simultaneously optimizing $z_b$ to $z_b$ via cosine loss, where $z_b$ is the  $h^{-1}$ output. The results proves that the model with  $h^{-1}$ (Fig2 \(c\)) is equivalent to SimSiam since it achieves comparable performance as the original SimSiam that directly optimizes $p_a$ approaching $z_b$.". Where are the "results"?
>
>
> The results are provided in Fig 3. (b) of the original paper and Fig 3. of the revised paper.
>
>
>
>
>
>
> ### The paper might have flaws in their proposed math. In subsec "Competition between o and r.", why "mo and mr is expected to be opposite of each other". The denominators of the two terms are both $||z||$. So only $||o_z||$ and $||r_a||$ define the values. However, $r_a = Z_a - o_z$. Thus $r_a$ and $o_z$ can have the same norms but with different directions? Why they need to be on the opposite directions?
>
> Sorry for the misleading sentence. We totally agree that this sentence does not always hold mathematically. In practice, we observe that they tend to have the opposite trend as shown in Fig. 4 of the revised paper. Nonetheless, we agree that the sentence in the original submission causes confusion and have fixed it.
>
>
> ### Sometimes it's hard to follow 3.1, 3.3 and 3.4
> The reply to all reviewers summarizes the changes made in the revised paper. For more detail, please check our revised submission. If anything remains unclear, please kindly let us know.

---

> ### Author Response · Authors · 2021-11-23
> **Second reply**
>
> > #### Summary of The Review: "I think the paper needs to improve descriptions about their framework and provide more experimental evidences."
>
> Thank you for the feedback. First, we have significantly improved the paper writing for describing our framework (see Sec 3 in the revised final paper). For more experimental evidences, we have summarized the results of various Siamese architectures in Table 2 with the training trend as well as the detailed setup reported in Appendix A.4. We have added the detailed setup for training SimSiam with Inverse Predictor in Appendix A.5.
>
> **Could you take the time to check our reply as well as the revised final paper? Thank you very much.**

---

### Official Review · Reviewer_q4XL · 2021-11-07

**Correctness:** 3
**Technical Novelty And Significance:** 2
**Empirical Novelty And Significance:** 2
**Recommendation:** 6
**Confidence:** 3

**Main Review:**

Pros:

1. The paper investigates the effects of center vectors and residual vectors in detail for both SimSiam and Info-NCE, which provides a unified explanation.

2. The results of SimSiam++ which shows a simple bias as a predictor can also avoid collapse without negative samples are interesting.


Cons:
 0. The writing of the paper needs to be improved. Many concepts are not explained or defined very clearly.

 1. The original SimSiam paper only claimed that "The usage of h may fill this gap (of missing EOA)." I think it's clear that the predictor does not learn to approximate the EOA. So I don't think the paper's interpretation of SimSiam's explanation is correct.

 2. In section 2.2, the paper claims that explicit EOA does not prevent collapse. But the experimental details are not explained very clearly here. I'm wondering whether the paper still uses one or two augmentations as the predictor's outputs or all the augmentations are used without stop-gradient.

 3. In section 2.3, the paper mentions "The results in Fig. 3(b)" show that it still leads to collapse. But I cannot find the collapsed results in Fig. 3(b). Besides, to prove Mirror SimSiam does not work (Fig 1. (c)), the authors should not apply stop-gradient to the predictor, because it's clear in the original SimSiam paper that fixed init does not work for the predictor. One possible way is to apply the gradient on z_a and p_b, and apply stop-gradient on z_b.

 4. In section 3.1, the paper mentions "Note that Z is l2-normalized, thus the trend of mo and mr is expected to be opposite of each other." This does not always hold.


Possible typos:

Section 2.3: Fig1 (c) and Fig2 (a) both lead to success ==> Fig1 (c) and Fig2 (a) both lead to failure

Section 3.1: loss - Z_a * sg(o_z) and loss - Z_a * sg(Z_a - o_z) ==> loss - Z_a \cdot sg(o_z) and loss - Z_a \cdot sg(Z_a - o_z)
Figure 3: - Z_a \cdot sg(Z_b - E(Z_b) ==> - Z_a \cdot sg(Z_b - E(Z_b))

Section 3.2: Z_n and r_n are used without definitions, which I guess means the representation for the negative examples.


**Summary Of The Paper:**

The paper proposes another explanation for why SimSiam can avoid collapse without negative samples. Specifically, the paper decomposes the gradient of learned representation as center vector and residual vector and finds that the center vector gradient has the de-centering effect and the residual gradient vector has the de-correlation effect. Such an explanation can also be applied to Info-NCE, which unifies the theory of self-supervised learning with and without negative samples.

**Summary Of The Review:**

Overall, I think the idea of analyzing the extra gradient component is novel. The writing quality of the paper is not very good.

---

> ### Author Response · Authors · 2021-11-19
> **First reply**
>
> Many thanks for the great comments! Please find our replies below.
>
> ### Cons 0.  The writing of the paper needs to be improved. Many concepts are not explained or defined very clearly.
> The reply to all reviewers summarizes the changes made in the revised paper. For more detail, please check our revised submission. If anything remains unclear, please kindly let us know.
>
>
>
> ### Cons 1. The original SimSiam paper only claimed that "The usage of h may fill this gap (of missing EOA)." I think it's clear that the predictor does not learn to approximate the EOA. So I don't think the paper's interpretation of SimSiam's explanation is correct.
>
>
>
> We totally agree that the predictor does not learn to approximate the EOA. However, we think that SimSiam's explanations may mislead the readers to believe the predictor learns to approximate the EOA. Here, we pick two sentences from the original SimSiam paper as follows.
> > *"But it may be possible for a neural network (e.g. the preditor $h$) to learn to predict the expectation".*
> > *"The usage of the predictor $h$ is presumably because the expectation $\mathbb{E}_{\mathcal{T}}[\cdot]$ in (9) is ignored".*
>
>
> Therefore in this paper, we hope to verify the point *"the predictor in SimSiam does not learn to approximate the EOA"* much more **explicitly with the experiments and analysis**.
>
>
>
> ### Cons 2. I'm wondering whether the paper still uses one or two augmentations as the predictor's outputs or all the augmentations are used without stop-gradient.
>
>
>
> In the EOA experiment, multiple augmentations, 10 augmentations for instance, are applied on the same image. With multi augmentations, we get the corresponding encoded representation, \ie $z_i$, $i\in [1,10]$. We minimize the cosine distance between the first representation $z_1$ and the average of the remaining vectors, *i.e.* $\bar{z} = \frac{1}{9} \sum_{i=2}^{10} z_i$. Gradient detach is applied on the average vector. Relevant content is added in Appendix A.3.
>
>
>
> ### Cons 3.1 In section 2.3, the paper mentions "The results in Fig. 3(b)" show that it still leads to collapse. But I cannot find the collapsed results in Fig. 3(b).
>
> Sorry for the missing result for Fig. 3(b). We have added relevant result in Table 2 with a full trend reported in Fig. 7 of Appendix A.4.
>
> ### Cons 3.2 Besides, to prove Mirror SimSiam does not work (Fig 1. \(c\)), the authors should not apply stop-gradient to the predictor, because it's clear in the original SimSiam paper that fixed init does not work for the predictor. One possible way is to apply the gradient on $z_a$ and $p_b$, and apply stop-gradient on $z_b$.
>
> We totally agree with your interpretation of Fig. 1\(c\). Actually, our experimental implementation indeed applies gradient on $Z_a$ and $P_b$. We thank you for pointing out this concern and have redrawn Figure 1 in the revised paper. The pseudo-code is provided in Appendix A.4.
>
> ### Cons 4. In section 3.1, the paper mentions "Note that Z is l2-normalized, thus the trend of mo and mr is expected to be opposite of each other." This does not always hold.
>
> Thanks for pointing out this concern. Indeed this does not always hold and we have fixed it and rewrote the relevant content in the revised paper.
>
>
>
> ### Cons 5. Typos
> Thanks very much and we have fixed the typos as suggested. We also go through the whole paper and try our best to remove other typoes.

---

> ### Author Response · Authors · 2021-11-23
> **Second reply**
>
> > #### Summary of The Review: "Overall, I think the idea of analyzing the extra gradient component is novel. The writing quality of the paper is not very good.
> Thanks again for recognizing the novelty of our work as well as suggesting improvement on the paper writing. We have performed multiple rounds of paper revision to improve the paper writing. We believe that the writing quality of the revised paper has been significantly improved.""
>
> **Could you take the time to check our reply as well as the revised final paper? Thank you very much.**

---

> > ### Comment · Reviewer_d9ds · 2021-11-29
> > **Re: Second reply**
> >
> > I appreciate the authors' efforts in addressing the raised problems. My several concerns are addressed so I decide to raise my score to 6.

---

> > > ### Author Response · Authors · 2021-11-29
> > > **Thanks for raising the score**
> > >
> > > We are happy that our effort in the rebuttal well addressed your concerns. It is very appreciated that you check our rebuttal and raise the score. Thank you very much!

---

> > ### Comment · Reviewer_q4XL · 2021-11-29
> > **Response to Authors**
> >
> > I appreciate the authors' efforts in rebuttal. After several times of revisions, the quality of the paper is much better than its first version, so I decide to raise my score to 6.

---

> > > ### Author Response · Authors · 2021-11-29
> > > **Thanks for raising the score**
> > >
> > > We thank Reviewer q4XL for the effort in checking our revised paper. We appreciate that you enjoy reading it with the comment "the quality of the paper is much better than its first version". Thank you very much!

---

### Official Review · Reviewer_TeJe · 2021-11-07

**Correctness:** 3
**Technical Novelty And Significance:** 3
**Empirical Novelty And Significance:** 3
**Recommendation:** 8
**Confidence:** 4

**Main Review:**

+ The topic of the research is of significance, as it is important to understand why designs like SimSiam/BYOL does not collapse. The paper attempts to get more insights of it with empirical evidence, and also shows the potential of connecting to contrastive methods (via InfoNCE loss). This is a great step toward better understanding of existing methods.
+ I like the style of providing conjectures and show results that empirically verifying them. The experimental designs are quite solid and insightful.

- Paper writing: I am a bit lost when reading the second part of the paper about the center vector and residual ones. I think it is good to have a clear definition for X in "centering w.r.t. X". Is X the entire image set and all possible augmentations? Is X just for the current image? Or is X for the current batch? Also I think this center vector must be approximate (e.g., taking multiple crops to get the average), not fully exact. So it would be good to have some notational clarifications on this. Particularly when introducing the analysis on negative examples and predictor, the notation becomes quite messy and hard to follow.
- About Fig 1 (c): if the stop gradient is applied this way, then the predictor is NOT going to be trained at all and would naturally leads to collapse to me. I think the better way is to have the stop gradient between the predictor and the encoder (so at the place of z_b). At least this way all the parameters in the framework are being trained. I do think even this placement will lead to collapse.
- For the part that revisits SimSiam's explanation: while it is great to have this part, to me it is not the essential part of the paper (the second part is). So I think it would be good to make the second part more self-contained, with richer set of experiments, and move the SimSiam explanation revisit part to appendix.
- Minor for the text that leads to Eq (2): I think symmetric loss is not the essential reason why SimSiam does not collapse, so it would be good to just focus on the asymmetric architecture (like SimSiam's teaser figure) and loss for the current analysis. It would also reduce the confusion.
- Experimentation wise, it is good that the current paper provides a good explanation, but performance wise the theoretical insights have not led to better results. It is less important for a paper like this, but calling the last section "SimSiam++" but without numbers/speeds that actually outperform SimSiam is strange to me.

**Summary Of The Paper:**

As the title suggests, the paper does a detailed investigation of how SimSiam avoids collapse without negative training examples. The key idea is to decompose the original vector into a center vector component and a residual vector component. The center vector cannot be too large (otherwise it is indicating collapse). The high-level idea is that the designs in SimSiam (and contrastive frameworks that have InfoNCE loss)  are mainly to prevent the center vector from getting too large. There are conjectures (verified empirically) about the relationship between the gradient w.r.t. the center vector ands the gradient w.r.t. the residual vector, and with these the paper finds that for SimSiam, the predictor is important for preventing collapse, particularly by doing de-correlation among features.

**Summary Of The Review:**

Overall I think this is a paper worthy of publication, with 1) its additional experiments to find flaws in the original hypothesis of SimSiam; and 2) explanations by decomposing the vector into two parts. However, I do have some concerns about the writing of the paper, so it would be good if the paper could have a higher quality (with proof-reading) and better organization so that it presents a more clear, unified message.

---

> ### Author Response · Authors · 2021-11-19
> **First reply**
>
> Many thanks for the great comments! Please find our replies below.
>
>
> ### 1. Clarification about the center vector and residual ones, and unclear notations in the second part of the paper.
>
> 1. Regarding center vector, by definition, ''centering w.r.t. X'' is the average of the entire image set and all possible augmentations. In practice, however, we adopt the representations from the current batch as an approximation. Empirically, we find that the batch-wise average of vectors is a good approximation if the batch size is large enough, 256 for instance. A sample-specific residual vector is calculated by $Z - o_z$.
>
>
> 2. When introducing the analysis on negative samples and predictor, the confusion on the notation probably comes from the extra gradient component. Recognizing this, we rewrite the part for introducing an extra gradient component in Sec 3.1, where we make the definition more clear in the context. For notation, since Z has been extensively used in other contexts, we change the notation from $Z_e$ to $G_e$. Moreover, we spent a large effort to make our paper more concise and restructured the content to make the logical link more clear. We hope the revised version clarifies the confusion. Please let us know if anything in the revised version remains unclear.
>
>
>
> ### 2. Stop gradient in the Fig 1 \(c\).
>
> 1. As indicated in original Fig. 1\(c\), we totally agree with you that the predictor is NOT going to be trained in the model. The stop gradient should be placed between the predictor and the encoder.
>
> 2. We highlight that our experimental implementation **indeed** puts the stop gradient between the predictor and the encoder to make all parameters in the framework trained. With this implementation, the predictor is trained in the same as that in the original SimSiam, while the gradient on the encoder does not backward through the predictor. We thank you for pointing out this concern and have redrawn Figure 1 in the revised paper. Pseudocode is provided in Appendix A.4.
>
>
>
> ### 3. First part of this paper.
>
> 1. Thank you very much for your great advice on the first part. Pointing out a reasoning flaw in a seminal work like SimSiam is a non-trivial contribution by itself and it is also the motivation for us to start this work in the first place. Thus, we believe that it is meaningful to keep it in the main body of this paper.
> 2. We fully agree with you that the second part is more important and thank you for your kind concern that it might lack enough space to be made clear. In order to save space for it, we move some content in the first part to Appendix A.2 and spent the effort to make the remaining content in the first part more concise.
>
>
>
>
> ### 4. Misleading text about Eq(2).
>
> We agree with you that indeed that symmetric loss itself does not prevent collapse. What prevents the collapse is the combination of predictor and stop gradient in the symmetric loss. Thanks for pointing out the misleading text, and we have modified it in the revised paper as follows:
> "SimSiam solves the collapse problem via predictor and stop gradient, based on which the encoder is optimized with a symmetric loss:"
>
> ### 5. Inappropriate section title  "SimSiam++".
>
> Sorry for the misleading section title. We have changed the section title to "Towards simple and explainable predictors" to avoid confusion in the revised paper.
>
>
> Overall, we really appreciate your effort and great advice on this paper. If anything remains unclear, please let us know and we will make every effort to fix them. Thank you.

---

> ### Author Response · Authors · 2021-11-23
> **Second reply**
>
>
> > #### Summary of The Review: "Overall I think this is a paper worthy of publication, with 1) its additional experiments to find flaws in the original hypothesis of SimSiam; and 2) explanations by decomposing the vector into two parts. However, I do have some concerns about the writing of the paper, so it would be good if the paper could have a higher quality (with proof-reading) and better organization so that it presents a more clear, unified message."
>
> Thank you again for recognizing the value of our work as well as suggesting improvement on the paper writing. We agree that better paper writing is beneficial for **presenting a more clear, unified message**. We have performed multiple rounds of paper revision for this purpose. We believe that we have significantly improved the paper writing in the revised paper.
>
> **Could you take the time to check our reply as well as the revised final paper? Thank you very much.**

---

### Author Response · Authors · 2021-11-19
**Paper revision summary**

Thank all reviewers for the insightful feedback. Overall, it seems that all reviewers recognize the significance of this work while some have concerns about the writing quality. We spent a large effort to revise the whole paper as the reviewers suggested. Here, we summarize the main changes we have made.

First, we have moved part of the content in Sec 2 to the Appendix and made the remaining part more concise. We have redrawn Fig. 1 and Fig. 2, especially for Fig. 1\(c\) and Fig. 2\(c\).

Second, we have added the detailed setup for Explicit EOA in Appendix A.3. We have added results of various architecture in Table 2 and provided detailed trend and setup in Appendix A.4. We have added the detailed setup for an inverse predictor in Appendix A.5. We have clarified the center vector more concisely.

Third, we have clearly defined and discussed basic gradient and extra gradient in the context for avoiding confusion. A major effort of this work is spent on analyzing how the center vector and residual vector in the extra gradient help avoid collapse (Note that the basic gradient in naive Siamese architecture leads to collapse). Since $Z$ has been used to indicate the vector representation. To avoid confusion, we change $Z_e$ to $G_e$.

Fourth, we have added solid derivation for the gradient of InfoNCE and propose a hypothesis that the de-correlation of InfoNCE arises from the gradient putting different weights on negative samples based on their similarity to the anchor. We verify this hypothesis through the analysis of gradient in Appendix A.7.

Fifth, we have performed a theoretical derivation for the predictor with a single bias layer in Appendix A.8. We derive that the bias vector roughly converges to $k o_z$ where k is a certain (dynamically changing) scalar vector and $o_z$ is the center vector. The derivation is empirically verified through the results in Table 6.

Last but not the least, we have restructured part of the content, especially in Sec 3.3 and 3.4 as well changed the outline of some content to make it more reader-friendly. Moreover, we fixed those small typos as identified by the reviewers.

In the coming few days, we will continue to update the paper to make it more explicit and clear. If anything remains unclear to the reviewers, please kindly let us know.

Overall, we express our genuine gratitude to all the reviewers for helping enhance this work to a new level.

---

### Author Response · Authors · 2021-11-20
**Paper revision summary (second-round)**

Thank all reviewers for their effort to help improve this work.

We have performed a second revision of our paper. We summarize the major changes compared with the first revision version.

First, We have restructured the content in Sec 3.3. Especially, we have put the analysis of SimSiam and MirrorSimSiam in different paragraphs for avoiding confusion, especially regarding the relevant notations. We have also restructured the remaining part of Sec 3.3 into two paragraphs with a clear message in each of them.

Second, we restructure some content from Sec 3.3 to 3.4.

Third, we have performed a thorough proofreading of the whole paper.

We hope the reviewers can check this new version. If any part in this new version is still unclear, please kindly let us know.

---

### Author Response · Authors · 2021-11-21
**Paper revision summary (third-round)**

Thank all reviewers for the effort to help improve this work.

We have performed the third revision of our paper to further improve the writing quality. We summarize the major changes compared with the last revision version.

First, We have restructured the content in Sec 3 by making "Extra gradient component for alleviating collapse" into an independent subsection, which makes the message in each subsection more clear.

Second, we have made "Towards a unified understanding of recent progress in SSL" into an independent section for highlighting its importance. In this new section, we have added the results of temperature analysis from the supplementary.

Third, we have moved the discussion on whether BN alleviates collapse into Appendix B, which is referenced when it is introduced in the introduction. We move this part to the Appendix because we believe that it is important for the paper completeness but it is not the main message for our work.

Fourth, we have carefully proofread every sentence of the content in the main body of our work.

We hope the reviewers can take the time to read this new version. If anything remains unclear, please kindly let us know.

---

### Author Response · Authors · 2021-11-23
**Paper revision summary (Fourth and final round)**

### Thank all reviewers for the effort to help improve the quality of this work. All reviewers seem to have recognized the significance and novelty of our work while some have concerns about the paper writing. We take the opportunity to perform multiple rounds of paper revision in this rebuttal. We hope the final revised version addresses all the concerns.

**The main idea and content of this final revised paper stay the same, and we mainly follow the advice of reviewers to improve the paper writing. Here, we summarize the major revision changes (in the whole rebuttal) as follows.**

#### (a) Content moved to the appendix
We have moved a small part of the content in Sec 2 to Appendix A.2 and moved the content for discussing the effect of BN to Appendix B.

#### (b) Figure redrawing
As suggested by **reviewer Teje and reviewer q4Xl**, We have redrawn Fig. 1 and Fig. 2, especially for Fig. 1\(c\) and Fig. 2\(c\).

#### \(c\) Clear definition and clarification
As suggested by **reviewer Teje**, we have clarified the center vector more clearly. We have clearly defined and discussed basic gradient and extra gradient components with a clear context for avoiding confusion. Given that the basic gradient in naive Siamese architecture leads to collapse, a major effort of this work is spent on analyzing how the center vector and residual vector in the extra gradient help avoid collapse. For notation, we change $Z_e$ to $G_e$ to indicate the extra gradient component.

#### (d) Results to backup the claims
As requested by **reviewer d9ds** we have summarized the results of various Siamese architectures in Table 2 with the training trend as well as the detailed setup reported in Appendix A.4. We have added the detailed setup for training SimSim with Inverse Predictor in Appendix A.5.

#### (e) De-correlation in InfoNCE
As suggested by **reviewer D1d5**, we have added solid derivation for the gradient of InfoNCE in Appendix A.7, and formed a hypothesis that the de-correlation of InfoNCE arises from the gradient putting different weights on negative samples based on their similarity to the anchor. We verify this hypothesis through the results in Fig. 7.

#### (f) Theoretical derivation of how a single bias layer prevents collapse
As suggested by **reviewer D1d5**, we have performed a theoretical derivation of how a single bias layer in the predictor prevents collapse in Appendix A.8. We derive that the bias vector roughly converges to $k o_z$ where $k$ is a certain (dynamically changing) scalar value and $o_z$ is the center vector. The derivation is empirically verified through the results in Table 6.

#### (g) Restructuring the section content
As suggested by multiple reviewers, we have restructured the content on analyzing SimSiam and InfoNCE into three sections (Sec 3, Sec 4 and Sec 5), each conveying a clear message. In Sec 3, we have made every sub-section more clear and logically connected.
- Sec 3.1, we interpret the collapse as a competition between center vector and residual vector, and propose as well verify Conjecture1.
- Sec 3.2, we discuss why basic gradient in a symmetric architecture cannot avoid collapse, and clearly define and discuss the extra gradient in the context.
- Sec 3.3, we analyze how the extra gradient of a random negative sample helps prevent collapse via analyzing its center vector and residual vector respectively.
- Sec 3.4, We decompose the extra gradient in SimSiam into its center vector and residual vector, respectively. We find that the center vector alleviates collapse through de-centering based on Conjecture1.
- Sec 3.5, we show that the residual vector in the extra gradient of SimSiam achieves dimensional de-correlation which also helps alleviate collapse.
- Sec 4: We have demonstrated that InfoNCE with negative samples also achieves the de-centering and de-correlation as in SimSiam. Especially for de-correlation, we have demonstrated how the residual vector in the extra gradient component achieves the de-correlation. Overall, this helps unify SimSiam and InfoNCE as well as bridges their gap with other SSL frameworks that perform explicit de-centering and de-correlation.
- Sec 5: Our understanding of how SimSiam works motivates us to simplify the predictor. Interestingly, we have found that a single bias layer can be sufficient for preventing collapse, which has been justified by a theoretical derivation in Appendix A.8.

#### (h) Position change for figures and tables
We have also changed the position of figures and tables to make it more reader-friendly.

#### (i) Paper proofreading
We have carefully proofread the whole paper including the Appendix.

Overall, we express our genuine gratitude to all the reviewers for helping enhance this work to a new level.

**Could we request the reviewers to take the time to check our reply as well as our revised final paper to confirm whether anything remains unclear? If there are any other concerns, please kindly let us know.**

---

### Author Response · Authors · 2021-12-01
**Final thanks to all reviewers and AC(s)**

As the discussion approaches the end, we take the opportunity to express our genuine gratitude to all reviewers. All your comments are very constructive for improving our work. We also thank the AC(s) for their effort in helping evaluate our work.

Thank you all very much.

---

### Decision · Program_Chairs · 2022-01-20

**Decision:**

Accept (Poster)

**Comment:**

This paper provide an explanation why contrastive learning methods like SimSiam avoid collapse without negative samples. As the authors claimed, this is indeed a timely work for understanding the recent success in self-supervised learning (SSL). The key idea in this submission is to decomposes the gradient into a center vector and residual vector which respectively correspond to de-centering and de-correlation. Such an explanation is interesting and novel. The empirical results are solid and convincing. During the rebuttal stage, the concerns from the reviewers are well resolved, and the writing of the new version is significantly better than the original one.